# Comprehensive marine substrate classification applied to Canada's Pacific shelf

**Edward J. Gregr** [1,2]*, **Dana R. Haggarty**[3,4], **Sarah C. Davies**[3], **Cole Fields**[5], **Joanne Lessard**[3]

**1** SciTech Environmental Consulting, Vancouver, British Columbia, Canada, **2** Institute for Resources, Environment, and Sustainability, University of British Columbia, Vancouver, British Columbia, Canada, **3** Fisheries and Oceans Canada, Pacific Biological Station, Nanaimo, British Columbia, Canada, **4** Department of Biology, University of Victoria, Victoria, British Columbia, Canada, **5** Fisheries and Oceans Canada, Institute of Ocean Sciences, Sidney, British Columbia, Canada

* ed@scitechconsulting.com

**Data Availability Statement:** The data and associated project code have been uploaded to GitHub. The url is https://github.com/ejgregr/substrate_model.

## Abstract

Maps of bottom type are essential to the management of marine resources and biodiversity because of their foundational role in characterizing species' habitats. They are also urgently needed as countries work to define marine protected areas. Current approaches are time consuming, focus largely on grain size, and tend to overlook shallow waters. Our random forest classification of almost 200,000 observations of bottom type is a timely alternative, providing maps of coastal substrate at a combination of resolution and extents not previously achieved. We correlated the observations with depth, depth-derivatives, and estimates of energy to predict marine substrate at 100 m resolution for Canada's Pacific shelf, a study area of over 135,000 km$^2$. We built five regional models with the same data at 20 m resolution. In addition to standard tests of model fit, we used three independent data sets to test model predictions. We also tested for regional, depth, and resolution effects. We guided our analysis by asking: 1) does weighting for prevalence improve model predictions? 2) does model resolution influence model performance? And 3) is model performance influenced by depth? All our models fit the build data well with true skill statistic (TSS) scores ranging from 0.56 to 0.64. Weighting models with class prevalence improved fit and the correspondence with known spatial features. Class-based metrics showed differences across both resolutions and spatial regions, indicating non-stationarity across these spatial categories. Predictive power was lower (TSS from 0.10 to 0.36) based on independent data evaluation. Model performance was also a function of depth and resolution, illustrating the challenge of accurately representing heterogeneity. Our work shows the value of regional analyses to assessing model stationarity and how independent data evaluation and the use of error metrics can improve understanding of model performance and sampling bias.

## Introduction

Coastal management depends on understanding how marine species are distributed. Species distributions are central to protected area design, vulnerability assessments, ecosystem-based

**Funding:** Dr. Gregr is the principal of SciTech Environmental Consulting. His funding for this study was provided by Fisheries and Oceans Canada on a contractual basis. The funder did not have any additional role in the study design, data collection and analysis, decision to publish, or preparation of the manuscript.

**Competing interests:** This funding does not alter our adherence to PLOS ONE policies on sharing data and materials. The specific roles of these authors are articulated in the 'author contributions' section.

fisheries management, and other marine spatial planning activities such as aquaculture siting and oil spill response. Species distribution (or habitat suitability) models predict these distributions by mapping relationships with environmental predictors. Here, we take on the challenge of mapping of marine substrates, a key determinant of habitat for benthic species [1, 2].

The emergence of multibeam (MB) echosounder acoustics in the early 2000s [3] revolutionized how oceanographers and marine geologists view the sea floor [e.g., 4]. Prior to MB swath scanners, sea floor characteristics (depth and substrate) were based on point observations. Today, in addition to supporting sub-meter bathymetric models, acoustic backscatter (BS) intensity, typically collected at the same time as the bathymetry, is being used to derive sediment classifications [e.g., 5].

The allure of acoustic data to estimate sediment composition is strong because swath sensors can cover large areas at high resolution and are increasingly efficient with depth. However, acoustic surveys of shallow coastal areas are expensive and time consuming, particularly for complex coastlines [6]. In Pacific Canada about 20% of the Exclusive Economic Zone has been mapped with MB acoustics, and these data are particularly lacking in shallow, intertidal waters (Peter Wills, Fisheries & Oceans Canada, Hydrographic Services, personal communication). BS classification also faces ongoing technical challenges including standardized signal calibration, data rectification, and ground-truthing [7], and including BS data in correlative substrate models may not be useful [see 8, 9]. Thus, unlike predictors of ocean dynamics or chemistry (available from remote sensing or ocean circulation models), ecologically relevant descriptions of bottom type continue to elude researchers due to both sampling and analytic challenges.

As countries continue to increase their protection of marine spaces [10, 11], effective management will require this knowledge gap to be filled efficiently and quantitatively [12]. Given that a suitable MB BS substrate layer for the Canadian Pacific shelf remains decades away, we used a random forest classification of available observations to develop spatial layers at two resolutions (20 m and 100 m). Random forest models are seen as reliable [e.g., 13] because of an insensitivity to overfitting the data [14, 15] and the ability to accommodate a variety of relationships between observations and predictors [14, 16]. We extended an earlier random forest model of rocky reefs [8] to include Mixed, Sand, and Mud substrates, and added energy predictors (i.e., wave fetch, bottom current, and tides) and additional bathymetric derivatives. Our four substrate classes were defined to reflect ecological function and to allow different data sources to be combined (see S1 Table in S1 File for descriptions of the substrate observations and Gregr et al. [6] for details on the class derivation). We used observations of bottom type from multiple sources allowing us to consider sample bias and independence. We evaluated model performance (both model fit and predictive power) across resolutions, geographic regions, and depths. We present our results using a collection of diverse and interpretable metrics.

## Objectives

Our main objective was to build a comprehensive, ecologically-relevant coastwide map of marine substrate to support predictions of quality habitat for benthic species, and other applications. The importance of such predictions to marine spatial planning makes timeliness an additional objective, and necessitates using the best available data. We approached this objective by building a suite of models extending from the high water line to the continental shelf. We defined classes to capture all substrate types (although with low class precision), making them more broadly ecologically relevant than grain size or single substrate models. Our methods are transparent and reproducible allowing refinement and updates as required–an advantage given ongoing data collection.

To assess the reliability of our models, we examined how class prevalence, sampling bias, model extents and resolution, and depth interact to influence model performance. Specifically, we asked the following questions:

1. Can weighting classes by observation prevalence improve model predictions?
The effect of class prevalence on classification models has been well described [e.g., 17, 18], and recent work [8] confirms the random forest algorithm favors the over-sampled class [19]. This challenge is also a significant area of research in the machine learning community [19], where well-balanced classes are encouraged. Finding little on this topic in the marine substrate classification literature, we tested the effect of class prevalence using two parallel sets of models with and without class-size weighting.

2. What are the effects of model extents and resolution?
Physical and ecological processes can differ across regions showing variable parameterization over space or time [20], but the detection and interpretation of such non-stationarity is rarely done. We were therefore interested in whether a single coastwide model would perform similarly across large, physiographically distinct regions.

3. Do our models perform differently by depth? And if so, are these differences correlated with model resolution?
Our collective experience based on over 50 years of surveying substrate on the BC Coast suggests that substrate heterogeneity (spatial variability) decreases with depth. We therefore predicted that higher resolution models would perform better than coarser resolution models in shallower waters.

## Challenges

**Ecological relevance.** To be relevant as a predictor for habitat suitability models, substrate classifications need to include the full range of substrate types to support the diversity of benthic organisms. Recently, automated, machine-learning approaches to BS classifications have been used to predict particle size [13, 21, 22] as part of the European nature information system (EUNIS) soft sediment class. Automated classifications integrating hard and soft substrates [e.g., 6, 21, 23] are less common in the literature, likely in part because more classes tends to decrease model fit [22, 24]. While methods are available to derive comprehensive classifications with many classes [4, 25], these are labor and data intensive and have thus only been applied to local extents. Fortunately, while a representation of all bottom types is necessary to maximize relevance, the number of classes need not be large, since habitat models and ecological analyses typically don't require and often cannot accommodate detailed classifications [e.g., 26]. We therefore limited our classification to four ecologically distinct classes: Rock, Mixed, Sand, and unconsolidated Mud. This has the advantage of allowing multiple sources of substrate observations to be combined [e.g., 6], and facilitates reproducibility compared to methods more closely tied to particular data types.

To be relevant as a habitat predictor, substrate classifications also need to be comprehensive across space—from the high water line to the shelf break. The exclusion of the coastal zone [commonly called the white strip because of the absence of data—6] is a chronic problem despite this being both the most productive region of the ocean and the most impacted by human activities [27, 28]. We addressed this challenge by including observations for the entire depth range, from the intertidal to the shelf edge.

**Resolution.** Developing relevant substrate maps is also challenged by high local substrate variability. When substrate varies at the scale of meters (a common feature, especially in

shallow waters), the spatial heterogeneity of a substrate grid (i.e., raster) will depend on the resolution used because each pixel assumes homogeneity. Thus, a 100 x 100 $m^2$ model will show less variability, and potentially a different distribution of substrate classes, than a 20 x 20 $m^2$ model of the same area because point observations must be aggregated to the target resolution. This aggregation can reduce class accuracy, and limits the representation of variability to a single resolution. This suggests that model performance will increase with model resolution when point observations are used for validation. This assertion is supported by contrasting the performance of recent random forest grain size models built at different resolutions [24, 29, 30]. We examine the question of resolution by comparing our 100 m models to our 20 m models, the finest resolution achievable across the large spatial extents of our study area.

**Observations and predictors.** Substrate observations tend to be spatially patchy and biased towards different bottom types and depths according to sampling method. For example, Lawrence et al. [12] described the challenge of observational sampling when hard substrates are covered with a veneer of soft sediment, while data collected for safe navigation is often limited to shallower waters. Predictors can contain both sampling errors (e.g., poorly reconciled bathymetric track lines) and edge-effects (e.g., bathymetric derivatives generated by excluding terrestrial elevations [8, 31]). While spatial artefacts in the predicted layer can help identify systematic errors in the predictors, errors in the dependent data are harder to identify. This makes the degree of contextual overlap between observational data sets, which determines their shared biases, particularly salient when testing predictive power. Understanding how biases in the data used to build models compare to the data used to test their predictive power can provide insight into the limits of model complexity and improve understanding of model performance and scaling [32].

**Model performance.** For acoustic data collected with remote sensors, tree-based classifiers such as random forest models are now the most common method applied [e.g., 13, 21]. Similar statistical methods are used in ecological studies to classify species observations into predictions of suitable habitat [e.g., 20, 33], and both applications rely on correlations with environmental predictors. However, because ecological observations are patchy, spatial predictions of habitat suitability rely on the continuous distribution of predictors to make habitat maps [20]. Thus, maps derived from observations rely on the strength and stationarity of the predicted relationships. This is why tests of predictive power (as opposed to simply model fit) are essential to evaluating point-based classifications of habitat suitability, and are adapted here.

Evaluating model performance requires appropriate metrics and testing data, and when maps are based on functional relationships (as in the case of point-based models), the consideration of process stationarity [a common but generally false assumption– 32, 34]. The application of performance metrics has evolved little in over twenty years of predictive modeling [35] with many studies continuing to report Cohen's Kappa as a measure of model quality despite its well-described shortcomings [36, 37]. While alternatives continue to appear in the literature [e.g., 38–40], adoption of these improved metrics has been slow, likely because papers with equations tend to be poorly cited by many practitioners [41].

There is also a persistent misconception about how to interpret model performance given the testing data. The majority of models are tested using cross-validation (the splitting of a set of observations into training and testing partitions) a process described as internal validation [42] or tests of model fit. To test model predictive power [35] (also called forecast skill [e.g., 32], external evaluation [42], or model transfer [43]), independent data are required [39, 43–45]. While independent data collected for purpose are desirable [e.g., 39], the use of opportunistic data can serve a similar purpose, while also illustrating important differences among data contexts [e.g., 32].

**Table 1. Contents of the build data set showing number of observations (total and by substrate class) available for model development (build) and independent model evaluation.**

| Role | Type | Source | N | Class | | | |
|---|---|---|---|---|---|---|---|
| | | | | *Rock* | *Mixed* | *Sand* | *Mud* |
| **Build** | Grab | CHS | 127,770 | 58,899 | 13,688 | 34,753 | 20,430 |
| | Grab | NRCan | 8,938 | 0 | 0 | 4,241 | 4,697 |
| | Dive | DFO | 44,809 | 21,250 | 7,626 | 13,464 | 2,469 |
| | ROV | DFO | 10,856 | 4073 | 2205 | 3200 | 1378 |
| | Marsh | CHS | 5,214 | 0 | 0 | 0 | 5,214 |
| | **Totals** | | 197,587 | 84,222 | 23,519 | 55,658 | 34,188 |
| **Evaluation** | Dive | DFO | 4974 | 2,892 | 543 | 974 | 565 |
| | Camera | DFO | 2143 | 421 | 491 | 654 | 577 |
| | ROV | DFO | 6064 | 1,477 | 1,479 | 633 | 2,475 |
| | **Totals** | | 13,181 | 4790 | 2513 | 2261 | 3617 |

Data types included Grab and Dive samples, observations from drop Cameras and remotely operated vehicles (ROV), and chart annotations of Marsh. Data were sourced from the Canadian Hydrographic Service (CHS), Natural Resources Canada (NRCan), and Fisheries and Oceans Canada (DFO).

Evaluating the effect of spatial sampling patterns on predictions based on aspatial correlative relationships [20] is also a significant challenge. Examining residuals is recommended for assessing spatial autocorrelation and model stationarity, but we found no guidance on using residuals to evaluate categorical predictions. Other work has examined the spatial variability of model error by calculating performance metrics at a number of small, randomly positioned sites [46]. This approach was not feasible for our study because we could not control for the effect of sampling density on model performance. Instead, we approached this challenge by testing for model stationarity across regions and depths.

## Methods

We applied a random forest classification [14, 16, 47] to a collection of substrate observations (Table 1, the build data) to create predictive models of our four substrate classes (Rock, Mixed, Sand, and Mud) based on a suite of geophysical predictors (Table 2). We built a 100 m (100 x

**Table 2. Predictors used to classify the observational data and the data sources from which they were derived for each study area.**

| Predictor | Study area | Source | Native resolution |
|---|---|---|---|
| Depth | Coastwide[B] | Carignan et al. [48] elevation model | 3 arc-seconds (~90 x 90 m$^2$) |
| Slope | | | |
| Slope (std. dev.) | | Gregr [49] elevation model | 100 x 100 m$^2$ |
| Curvature | Regional | Davies et al. [31] elevation models | 20 x 20 m$^2$ |
| Rugosity | | | |
| Broad BPI[A] | | | |
| Medium BPI | | | |
| Fine BPI | | | |
| Tidal speed | Coastwide Regional | Mean summer conditions averaged from a regional circulation model [50]. | 3 x 3 km$^2$ |
| Ocean circulation | Regional (SOG only) | Mean summer conditions averaged from a local circulation model [51]. | 440 x 500 m$^2$ |
| Fetch | Regional | Sum of fetch based on Gregr [52]. | 50 m |

A. Benthic Positioning Index. See S2 Table in S1 File for details.

B. The two Coastwide elevation models were combined by Nephin et al. [33] into a single 100 m bathymetry from which the derivatives were calculated.

Each predictor was generated for both the Coastwide and Regional study areas at their respective resolutions, but often from different source data.

100 m$^2$) coastwide model for the Canadian Pacific continental shelf. To improve the quality of habitat models for species close to shore and extend the predictions across the white strip, we then built 5 regional, nearshore models at 20 m resolution (20 x 20 m$^2$) within the extents of the coastwide model (Fig 1). Our regions included the sheltered, largely muddy Strait of Georgia (SOG), the exposed West Coast of Vancouver Island (WCVI), the oceanographically distinct islands of Haida Gwaii (HG), the North Central Coast (NCC) with its deep fjords and inlets and a large exposed coastline, and the transitional Queen Charlotte Strait (QCS) region containing a mix of sheltered and exposed areas, fjords and inlets.

For each of the 6 extents we built models with and without class weighting to test for the effect of class prevalence. This collection of models allowed us to examine the relative performance of models across resolutions, regions, and depths.

We tested model fit by partitioning the build data into training and testing partitions. We tested the predictive power of our models using three independent data sets, collected separately. Our predictor variables (Table 2) included a commonly used suite of geomorphic predictors derived from bathymetry, and several measures of energy. Each of these data sets is described in the following sections.

### Model build data

We assembled a coast-wide data set of 197,587 observations from Natural Resources Canada (NRCan), the Canadian Hydrographic Service (CHS) and Fisheries and Oceans Canada (DFO) data holdings to build the models (Table 1). The observations are broadly distributed across Canada's Pacific shelf with high concentrations of points near shore (S1 Fig in S1 File). CHS collects grab samples as part of their regular hydrographic surveys and represent the largest component of our build data set, with sampling biased towards shallow waters and rocky substrate because of the CHS's mandate to chart navigable waterways. We therefore included NRCan grab data and marsh locations mapped by CHS to increase the prevalence of the soft bottom type classes in our build data. NRCan core samples are biased towards unconsolidated substrates. Direct observations of bottom type were acquired from DFO shellfish stock assessment dive surveys and remotely operated vehicle (ROV) surveys of rockfish habitat. Observations were re-classified to the four bottom type classes used in this analysis following Gregr et al. [6]. Details on these data are provided in (S1 Table in S1 File).

### Predictor data

Environmental predictors were selected to include a combination of benthic terrain features derived from bathymetry [e.g., 8, 53] and measures of energy [e.g., 33, 54]. To support our analysis across two spatial resolutions, we derived the same terrain features from a 100 m coastwide bathymetry and our 20 m regional bathymetries. These included slope, curvature, rugosity, the standard deviation of slope, and three bathymetric positioning indices (BPIs) with increasing neighborhood sizes (S2 Table in S1 File) to capture both small benthic features, and larger trends in terrain. Energy was represented using tidal currents and broad-scale circulations derived from ocean current models [50, 51]. Fetch, a proxy for wind-wave exposure [55], was included in the 20 m model but not the 100 m model as the shallowest accurate prediction from the 100 m was expected to be deeper than shoaling depth. Additional details on the derivation of these predictors is provided in (S2 Table in S1 File). We did not explore questions of predictor independence or variable selection.

### Independent evaluation data

Our independent data were collected at random locations by DFO as part of the Benthic Habitat Mapping Project, intended to define nearshore habitat & species assemblages. Surveys were

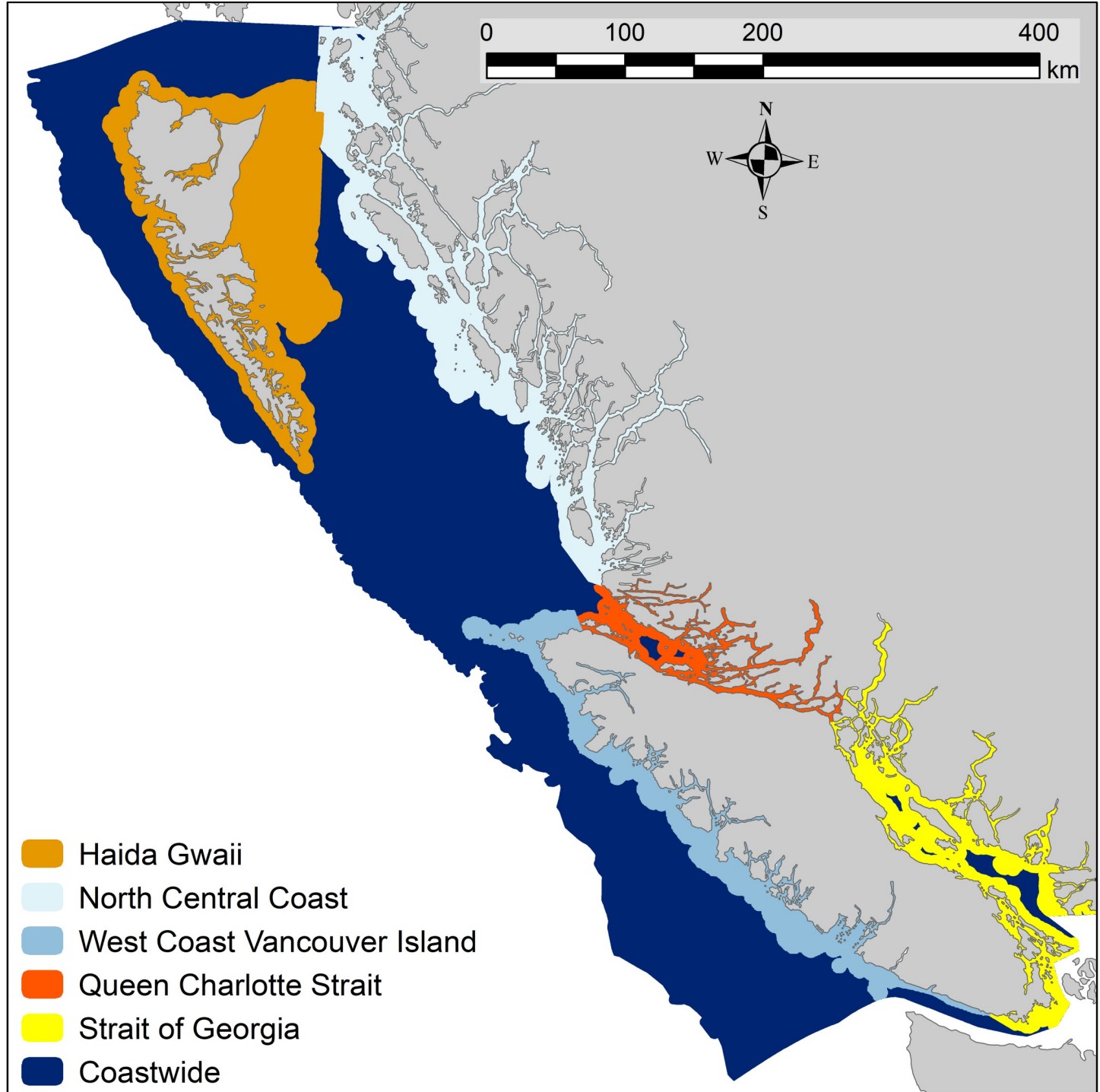

**Fig 1. Study area.** The spatial extents of the six models developed in this analysis. The 100 m coastwide model covers the Canadian Pacific continental shelf. The five 20 m regional models extend from the high water line to as far as 5 km seaward from the 50 m depth contour, the limit of the bathymetry and derived predictor variables. See text for details.

done using dive transects, drop cameras, and remotely operated video (ROV). The dive and ROV data were collected using similar methods as the build data, but they were collected at different times and often by different observers. The datasets included depth and consistently

coded substrate classes making it easy to reclassify them to our four substrate classes (S3 Table in S1 File). Observations were collected within quadrats on transects. We aggregated the observations from each quadrat by 20 m grid cells, assigning the mode substrate observation to the center point of a cell. These observation points were then used to extract predictor data from the 20 m and 100 m raster stacks.

Independent observations from the dive data included 1077 transects surveyed between 2013 and 2018. Data were collected following Davies et al. [56] in shallow areas along the coast, ranging from -5 to 19 m depth. Points above chart datum were surveyed by divers at a high tide. Data were prepared in the same way as the dive component of the Build data (S1 Table in S1 File).

Drop camera observations were collected following Davies et al. [56] using a GoPro camera deployed off the side of small boats during the dive surveys (2014 to 2018). We obtained still photos from 889 locations at depths from 16 to 60 m. These data were intended to extend the dive observations of substrate into deeper waters. Uncertainty in the positional accuracy of the images increases with depth due to deflection of the drop camera from the boat position. We minimized this uncertainty by removing locations where the difference between recorded depth and bathymetry exceeded 50 m.

ROV observations for Haida Gwaii and the North Central Coast regions were extracted from video imagery along 366 transects in depths from 33 to 675 m collected between 2013 and 2015. Observations were recorded for 10 second increments of video and aggregated by mode into the 20 m bins as described above.

## Model development and comparisons

All observational and predictor data were prepared using ArcGIS [57] before importing into R, where we joined all observational data with predictors scaled to coastwide (100 x 100 m$^2$) and regional (20 x 20 m$^2$) grids (i.e., rasters) before analysis. Predictors were not transformed or assessed for correlation since the random forest approach is largely robust to non-normal, correlated predictors [14, 16]. Where multiple observations occurred in the same raster cell, the predictor values were duplicated to preserve the observational sample size. This was more common for the coastwide model.

We built our models using the ranger package [58]. While a variety of random forest packages are available, ranger is the only one to effectively support the weighting of classes based on prevalence. We used the same number of trees (1000) and test fraction (0.6) for all models, set variable importance to use the Gini index, and the internal cross-validation to sample with replacement. All analyses were done in R [59], using a number of packages for the data analysis and presentation of results (see S2 Analytical Methods in S1 File).

We randomly spilt the build data into training (67%) and testing (33%) partitions. We used the training partition to build the models and the testing partition to assess model fit and explore the effects of depth and model resolution. We tested all models using the same testing partition to provide a baseline for assessing how model predictive power (based on IDE) may be influenced by sampling bias. We compared the weighted and non-weighted models to illustrate the effects of class prevalence. We did not use cross-validation as this is done internally as part of the random forest process, and is reflected in the out of bag error (OOB) [16]. We used the independent data to examine sample bias and test for stationarity. We evaluated model performance using a suite of comprehensive and interpretable metrics (see next section).

We weighted classes according to their prevalence in the training data ($1 - N_{class} / N_{total}$) and compared class performance using both the build testing sample and the independent data sets

across all models. We tested whether any observed differences were influenced by model resolution, region and depth.

To test for stationarity, we compared how the coastwide model performed against each of five regions. We tested our hypothesis that model performance is correlated with depth by comparing model fit and predictive performance across depth classes. We classified depths following Gregr et al. [6], who provided an ecological rationale for dividing coastal waters into Intertidal, 0–5 m, 5–10 m, 10–20 m, and 20–50 m zones. To these we added three deeper zones (50–100 m, 100–200 m, and 200+ m) for a more complete comparison of depths.

### Measuring model performance

Best practice now calls for using more than one metric [44], as no single accuracy metric can serve all assessment objectives, and different measures can imply different conclusions [60]. To ensure our assessments of model performance were comprehensive, we explored metrics developed across the disciplines of image classification [61, 62] habitat suitability modelling [35, 42], and weather forecasting [63, 64]. We report metrics describing both model accuracy and model error. Accuracy measures both how much better model predictions are than a random guess [42], and the observed agreement between predictions and a test dataset [62]. For better-than-random, we used the True Skill Statistic (TSS) instead of Kappa, which has been shown to have limited utility as a performance metric [36, 37]. We used Overall Accuracy [61] and True Negative Rate [TNR, 42] to provide information on correctly predicted positives and negatives respectively (aggregated across classes). We assessed by class accuracy using TNR, and User and Producer accuracies (see S1 File). We used measures of model error based on the work of Pontius and colleagues [e.g., 62]. These include Quantity error, which measures the deviance in the frequency of observations and predictions, Exchange error defined as a swapping between two categories, and Shift error, the remaining error that cannot be attributed to either Exchange or Quantity. We report these error metrics aggregated across classes. Our combination of accuracy and error assessment provides a more complete picture of model performance than is commonly reported. Finally, we derived Imbalance as an integrated measure of prevalence in a multi-class data set (see S1 File).

To complement the quantitative assessment, we examined the spatial agreement of our predictions with two well-known areas of the Pacific coast, Pacific Rim National Park Reserve in the WCVI region, and English Bay, part of the urban coast of Greater Vancouver in the SOG region. This qualitative comparison adds valuable information on how location influences the assessment of the substrate predictions by allowing the patterns produced by different models to be compared.

## Results

### Model development and class weighting

All models, whether weighted or not, had comparable and high fit to data (TSS values ranging from 0.56 to 0.64) with no notable differences between the coastwide and the regional models in the aggregated metrics (Table 3). The effect of weighting is more apparent in the error assessment where the Quantity errors of all non-weighted models (0.07 to 0.10) were about twice that of the corresponding weighted models (0.03 to 0.06) (Table 3). The majority of model error came from Exchange between classes. The reduction in Quantity error achieved by weighting tended to be offset by a corresponding increase in Exchange error. This explains why the aggregate metrics TSS and Accuracy were largely unchanged by weighting (Table 3).

Across classes, Rock and Mud had the highest User and Producer accuracies and the Mixed class had the lowest across all models, regardless of weighting (Fig 2). The effect of weighting

**Table 3. Aggregated build metrics for all 6 models comparing the weighted (first row) and the unweighted (second row) random forest results.**

| Model | N | Imbalance | OOB | TSS | Accuracy | TNR | Quantity | Exchange | Shift |
|---|---|---|---|---|---|---|---|---|---|
| Coast | 66056 | 0.17 | 0.30 | 0.57 | 0.70 | 0.86 | 0.03 | 0.25 | 0.02 |
|  |  |  | *0.29* | *0.59* | *0.70* | *0.85* | *0.07* | *0.20* | *0.02* |
| HG | 9191 | 0.17 | 0.26 | 0.64 | 0.74 | 0.88 | 0.06 | 0.19 | 0.02 |
|  |  |  | *0.25* | *0.64* | *0.74* | *0.87* | *0.08* | *0.17* | *0.01* |
| NCC | 22189 | 0.22 | 0.28 | 0.57 | 0.72 | 0.83 | 0.05 | 0.21 | 0.02 |
|  |  |  | *0.28* | *0.58* | *0.72* | *0.80* | *0.10* | *0.16* | *0.02* |
| QCS | 4383 | 0.20 | 0.29 | 0.56 | 0.70 | 0.82 | 0.06 | 0.22 | 0.02 |
|  |  |  | *0.29* | *0.57* | *0.71* | *0.79* | *0.11* | *0.16* | *0.02* |
| SOG | 14399 | 0.15 | 0.30 | 0.60 | 0.71 | 0.87 | 0.06 | 0.23 | 0.01 |
|  |  |  | *0.30* | *0.60* | *0.71* | *0.86* | *0.08* | *0.20* | *0.02* |
| WCVI | 9523 | 0.22 | 0.24 | 0.63 | 0.76 | 0.87 | 0.03 | 0.21 | 0.00 |
|  |  |  | *0.24* | *0.64* | *0.76* | *0.83* | *0.08* | *0.16* | *0.01* |

Sample size (N) and Imbalance characterize the observational data. Out of Bag (OOB) values show the mean prediction error from the random forest internal cross-validation. The True Skill Statistic (TSS) measures how model performance exceeds random after correcting for chance and prevalence. Overall Accuracy, True Negative Rate (TNR), Quantity, Exchange and Shift provide an assessment of model error. See (S1 File) for details on the metrics.

can be seen when the class-based metrics are compared: weighting shifted User Accuracy away from Rock to the other classes, particularly Mixed and Mud, for all models (Fig 2). There was also a corresponding increase in the Producer Accuracy of the Rock class with weighting at the expense of the other classes, though this was more variable across models.

The reduced Quantity error from class weighting is also evident when the prevalence of the build testing partition was compared to model predictions (Fig 3). The over-prediction of the Rock class by the non-weighted models and the consistent under-prediction of the Mixed class were clearly mitigated by class weighting, which also aligned the prevalence of the Sand and Mud classes more closely to the observed values. More importantly, these changes in prediction prevalence were evident in small but significant shifts in the spatial distribution of the classes, with the weighted model producing a pattern that more accurately reflected known nearshore substrate in our two test locations (Fig 4). The predictions from the weighted model produced less Rock and more Sand on known beaches in Pacific Rim. Changes in English Bay were less apparent, but a shift from Rock to Mixed is evident. In deeper waters of Pacific Rim, the weighted model predicted less Sand and more Mixed substrates, although bathymetric artefacts were enhanced.

Variable importance (Fig 5) differed notably across models suggesting the dominant processes differed across regions. Our indicators of ocean dynamics (circulation and tidal) were the top two predictors for the coastwide model followed by bathymetry. Slope, broad- and medium-BPI, and the standard deviation of slope provided almost equal contributions. In contrast bathymetry was the dominant predictor for all regional models except HG, where bathymetry was second to fetch. Fetch was also important in the WCVI region, the other region we presumed would be strongly influenced by exposure. Fetch, broad-BPI, and tidal flow rounded out the top four variables for the regions. Despite such rankings the contributions of predictors can be very similar, particularly among those contributing least. For example in the NCC region, the three least influential predictors have virtually equal model contribution scores (Fig 5).

## Model resolution

The most significant predictors for the 100 m coastwide model were related to ocean energy with a native resolution of three km. In contrast, the regional 20 m models were closely tied to

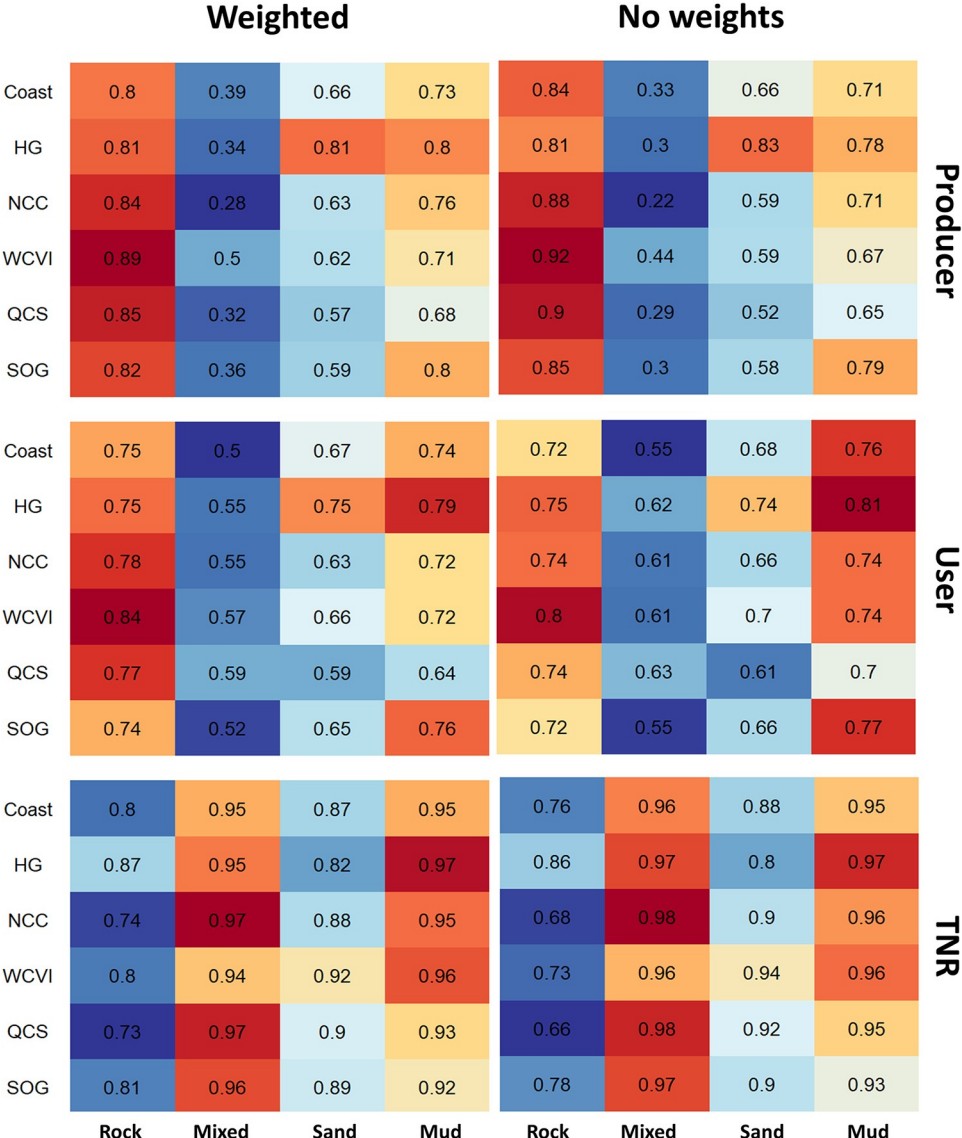

**Fig 2. Heat maps of model performance.** Producer Accuracy, User Accuracy, and True Negative Rate (TNR) for each substrate class, shown for weighted (left column) and non-weighted (right column) models. The color shading within each row reflects the underlying values from high (red) to low (blue) and is included to emphasize differences.

bathymetry and potential wave energy. Yet comparing the 100 m and 20 m models shows little difference in either the aggregated or class-based metrics between resolutions (Table 3 and Fig 2 respectively). However the differences in mapped predictions are dramatic, with the 100 m model showing an over-prediction of nearshore Rock in both focal areas despite class weighting (Fig 4). The mapped predictions from the 100 m model are also more homogeneous than the 20 m models (Fig 4). While this reduced visible artefacts, it also highlights the inability of the coarser model to represent local substrate heterogeneity.

## Performance across depths

The 100 m model shows a clear trend of increasing TSS with deeper water compared to the 20 m models based on the testing partition, although results were variable across regions

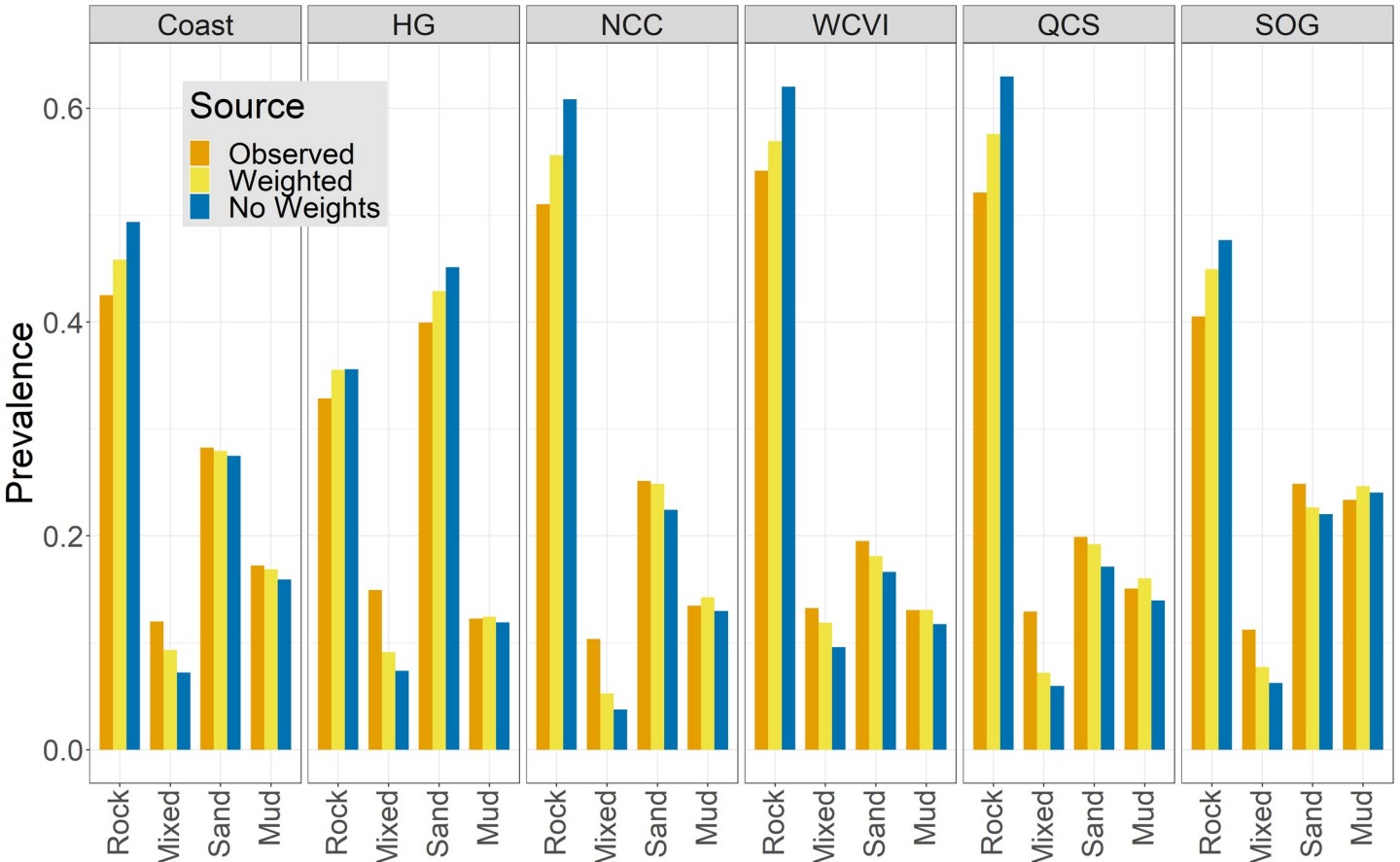

**Fig 3. Comparison of class prevalence.** Observed class prevalence in the build testing partition (orange) compared to predictions from the weighted (yellow) and unweighted (blue) random forest models across regions. Weighting tends to yield class prevalence closer to that observed in the build data (note that training and testing partitions have the same prevalence).

(Fig 6A). Specifically, the trend is clear for HG and NCC, absent for SOG, and uneven for WCVI and QCS. This pattern is driven by the higher Accuracy of the 20 m models in the intertidal and 0–5 m depth zones (Fig 7A) across all regions (except the SOG intertidal). The 100 m model has a consistently higher TNR, particularly in the 0–5 m depth zone (Fig 6B). The corresponding error assessment for the two resolutions also shows decreasing error with depth for the 100 m model across most regions (Fig 7A) but generally increasing with depth for the regional models (Fig 7B). All models show a tendency towards increased Quantity error with depth but most of the error is from an Exchange between classes.

The role of resolution in the correlation between model performance and depth is further supported by the IDE (see below) and is also apparent in the mapped predictions (Figs 4 and 8). The tendency of the 100 m model to predict more contiguous classes and over-predict Rock near shore is evident in all three of our test regions (Figs 4E, 4F and 8B). However, the 100 m model also captures known physiographic features in deeper waters, in particular the canyons in Queen Charlotte Sound and the shelf edge not identified by the 20 m models (Fig 8).

### Independent data evaluation

The independent data were not consistently distributed across regions: the Dive data were distributed most broadly, while the ROV data were limited to HG and NCC (Table 4 and S2 Fig

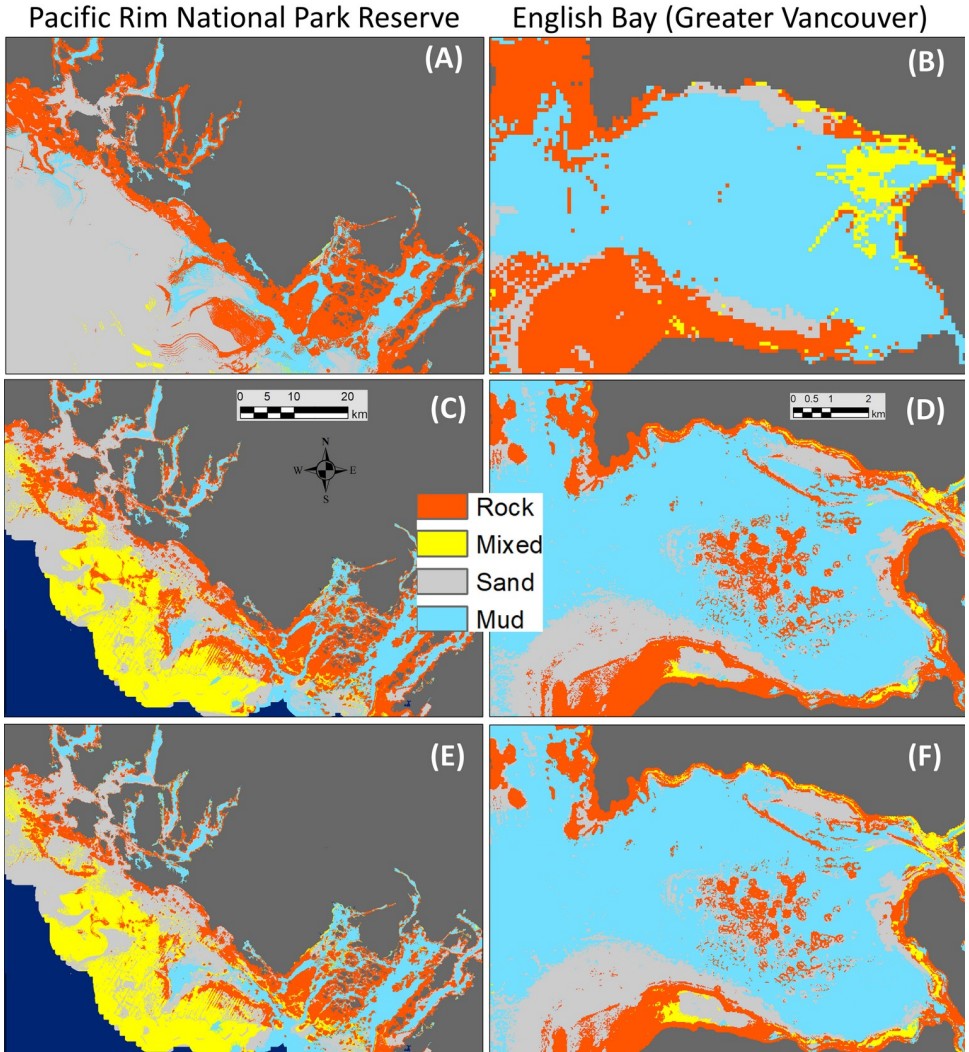

**Fig 4. Predictions in regional assessment areas.** Predictions from the 100 m coastwide (top row), 20 m, no-weight (middle row), and 20 m weighted (bottom row) models for the Pacific Rim National Park (left column) and Greater Vancouver (right column) assessment areas. The series of images (from top to bottom) shows how increased resolution and weighting for prevalence help mitigate the bias towards rocky substrate.

in S1 File). Not unexpectedly, model forecast skill was generally lower and more variable than model fit. The TSS for the 100 m coastwide model varied considerably (0.10 to 0.24) for the three independent data sets (Table 4), however all regional models had higher TSS scores than their corresponding coastwide model. The regional TSS scores were notably better for the Dive and ROV data. The pattern in Accuracy scores generally followed the TSS scores with some exceptions, showing the importance of accounting for chance in model performance. The TNR scores were highest for the ROV data followed by the Camera data, and notably lower for the Dive data (Table 4). Errors (Quantity, Exchange, and Shift) were highly variable across both regions and independent data sets.

Overall, the Dive and Camera data were predicted with the highest and lowest Accuracy respectively. Accuracy, while variable across regions, was inversely correlated with Imbalance (Table 4). The lower TNR predicted at the Dive observations (compared to the other

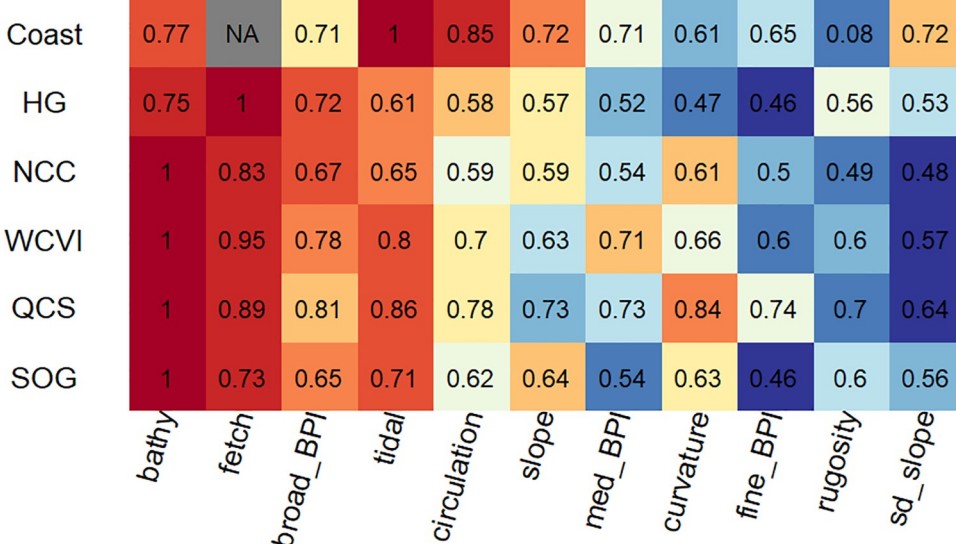

**Fig 5. Heat map of variable importance across models.** Variable importance, defined as the proportion of each predictor's contribution to the model, is shown relative to the predictor with the highest contribution ($p/p_{max}$). The color-shading within each row reflects the underlying numbers from high (red) to low (blue) and is included to make the differences in the values more apparent. For the coastwide model,"NA" indicates that Fetch was not used as a predictor.

independent data) is reflected in a correspondingly lower Quantity error (Table 4) which dominates the error component of most models, showing that much of the misclassification is due to errors in prediction prevalence.

Examining the aggregated scores adjusted to their no-information baselines (Fig 9) provided a clearer representation of relative predictive power. The regional models generally outperformed the coastwide model for regions (HG, NCC) with larger samples of independent data. For the other regions the models predicted the Dive data better than the coastwide model, but accurate predictions of the Camera data were variable. The ROV data (limited to the HG and NCC regions) were predicted with the most consistent Accuracy and TNR scores while the Dive data were consistently predicted with both the highest Accuracy and lowest TNR. These differences are due in part to sample size, but may also reflect some spatial bias in the sampling.

The influence of resolution on the correlation between depth and model performance is also apparent when the predictive power of the coastwide 100 m model is compared to the 20 m models for the regions with sufficient independent data (Fig 10). The IDE of the coastwide model (Fig 10A) shows a clear increase in predictive power with depth for both the Dive and Camera data, while the opposite pattern is evident in both the HG and NCC regions (Fig 10B and 10C). The IDE of the ROV data are more equivocal, illustrating differences between independent data sets and emphasizing the need to understand the different data collection methods and biases.

## Discussion

Our results show how class weighting to address sample imbalance can lead to both numerical and spatial improvements in model performance. We also confirm the existence of regional non-stationarity, and show that model reliability depends on depth, resolution, and substrate

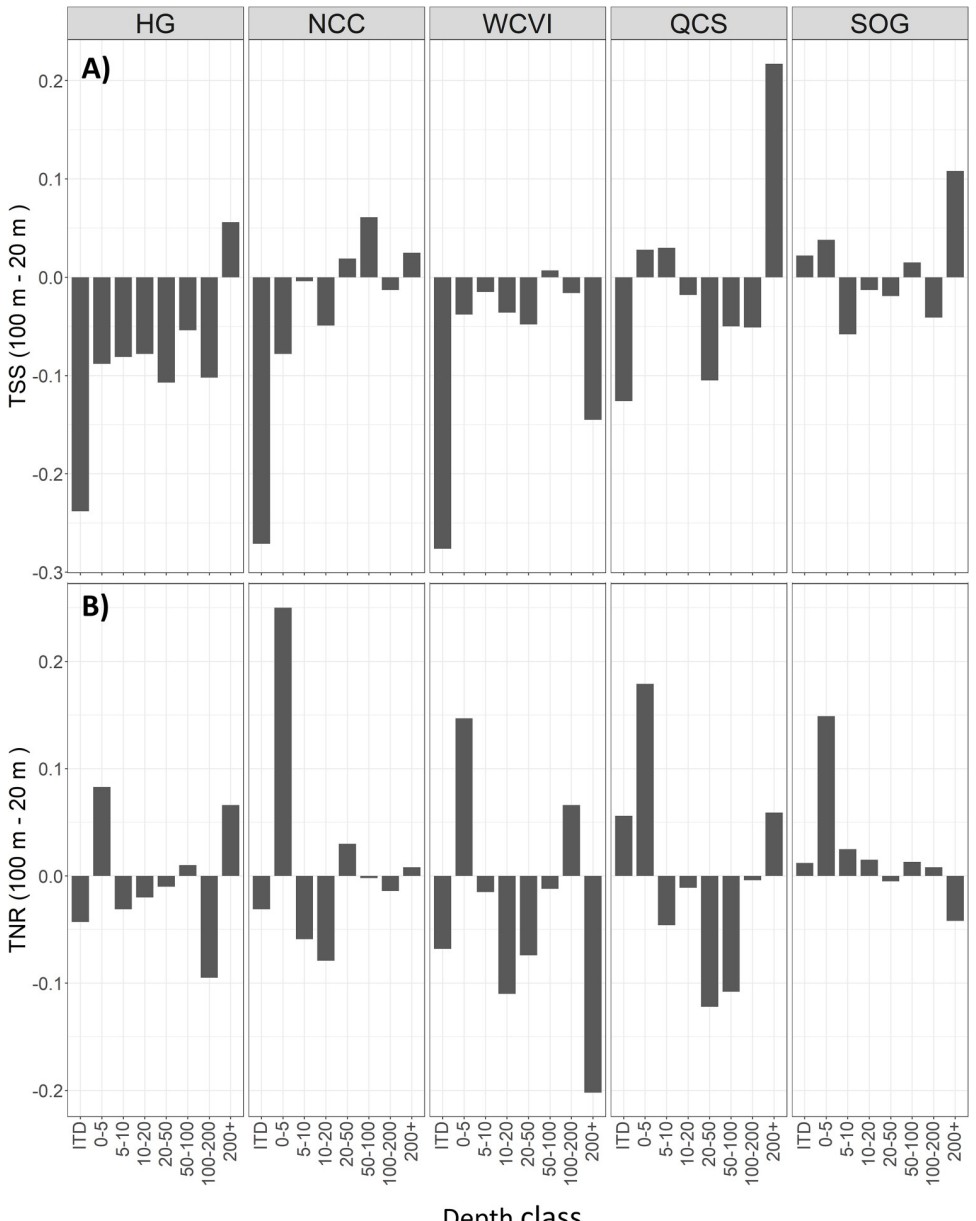

**Fig 6. Differences in model fit by region, across depths and resolutions.** The difference in (A) the True Skill Statistic (TSS) and (B) the True Negative Rate (TNR) between the 20 and 100 m models across depth ranges, shown by region. Scores are based on model fit to the build testing partition. Values below 0 indicate a higher score by the 20 m model. This shows how regional model performance is generally better across all depths and regions, except for SOG, and identifies possible sampling biases in the 0–5 and 200+ depth ranges.

class, and potentially the uniqueness (in terms of predictors) of the location in question. This means reliability will vary across the coast, with different resolutions and substrate classes being relatively more or less reliable in different locations. Understanding these differences will improve the confidence that can be placed in these and similar models. It will also inform their contribution to predictions of habitat suitability, and help guide future data collection and model refinement.

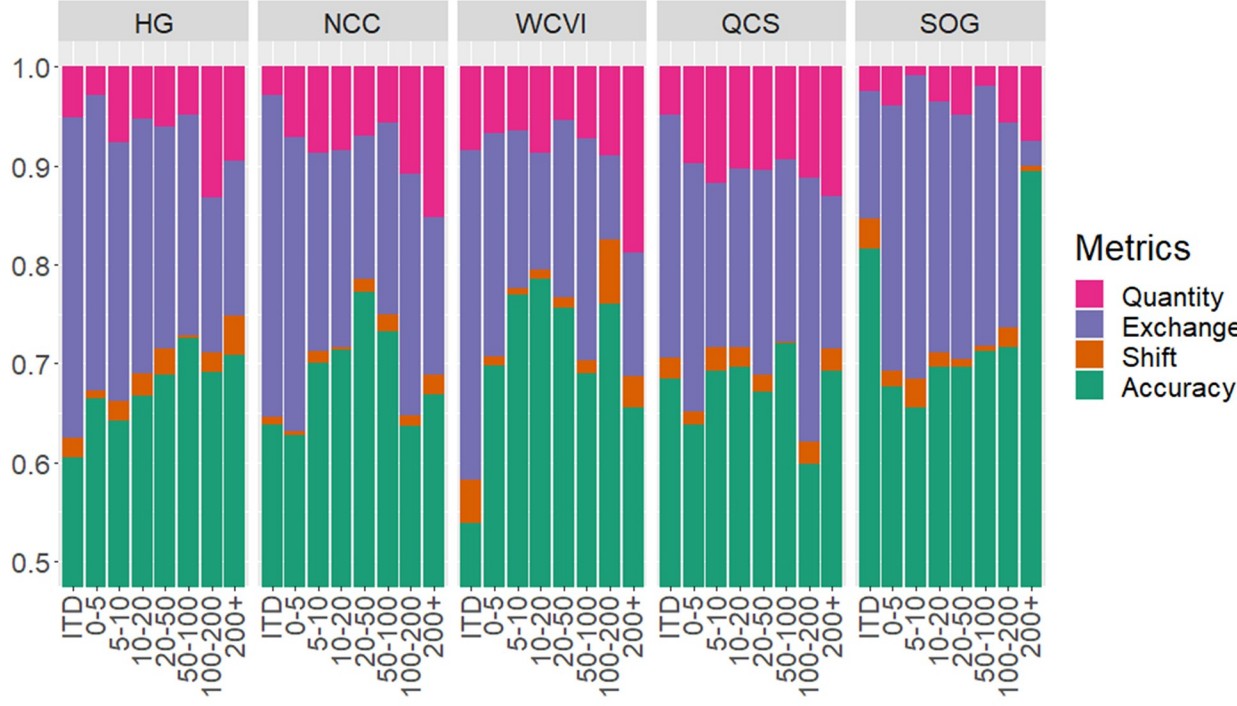

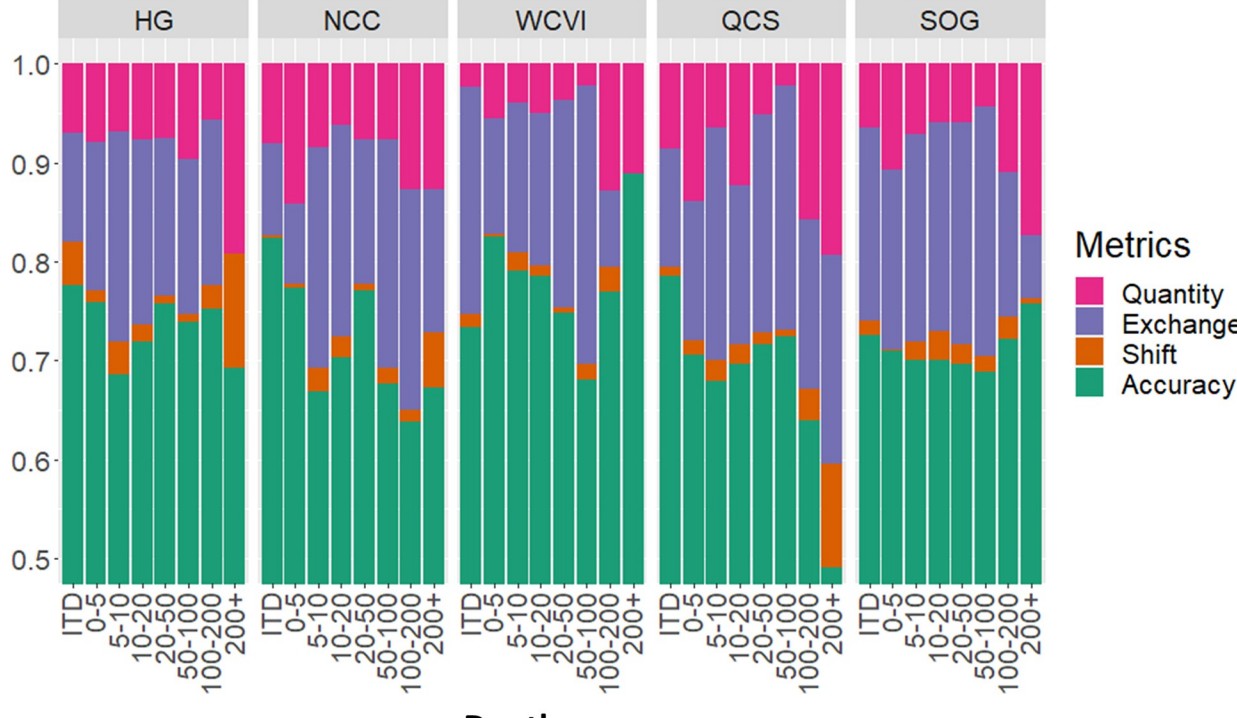

**Fig 7. Error assessment of model fit by region, depth and resolution.** Accuracy and error metrics for (A) the 100 m coastwide and (B) 20 m regional models shown across depth zones for each region, based on model fit to the build testing partition. Accuracy tends to increase with depth in the 100 m model and decrease with depth in the 20 m models, but the trends are noisy.

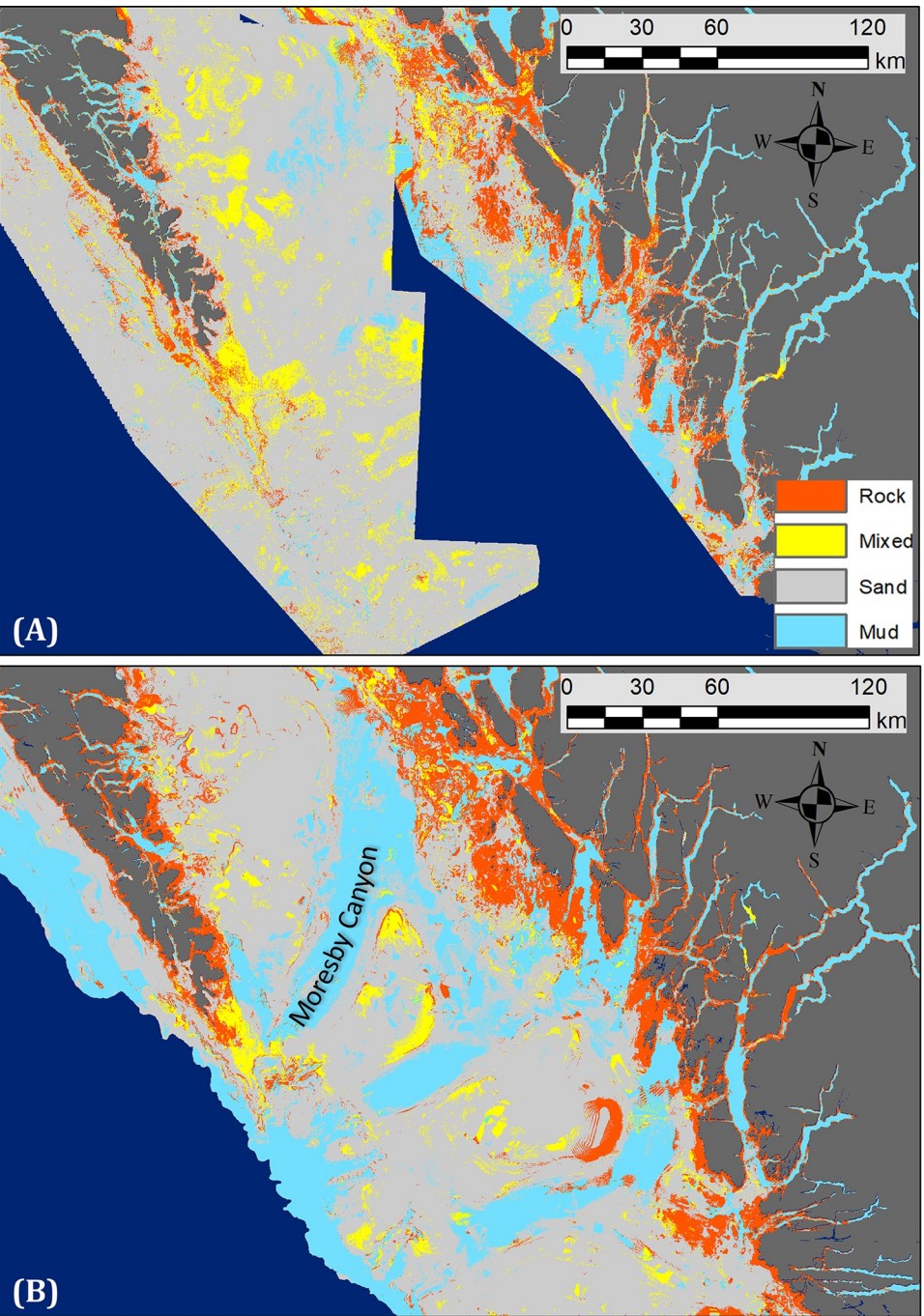

**Fig 8. Mapped predictions for different resolutions.** Predictions for portions of the HG and NCC 20 m regional models (A) are shown for comparison with the 100 m coastwide model (B). Note the detail provided by the 20 m models near shore where the 100 m model predicts largely Rock. In contrast, the 100 m model identifies known features at depth (e.g., Moresby Canyon) not captured by the 20 m models. The regional models are shown using unclipped predictor variables to allow model comparison in deeper waters. See text for details.

## Depth and resolution

All our models fit the build data well, however the predictive power of the regional 20 m models was universally better than the coastwide model. The consistently higher TNR of the 100 m

**Table 4. Performance of each random forest model against each independent data set (IDS) (see Table 3 for description of metrics).**

| IDS | Model | N | Imbalance | TSS | Accuracy | TNR | Quantity | Exchange | Shift |
|---|---|---|---|---|---|---|---|---|---|
| Dive | Coast | 3666 | 0.22 | 0.20 | 0.52 | 0.57 | 0.32 | 0.14 | 0.02 |
| | HG | 1479 | 0.23 | 0.32 | 0.58 | 0.75 | 0.08 | 0.31 | 0.03 |
| | NCC | 2217 | 0.30 | 0.33 | 0.65 | 0.57 | 0.19 | 0.14 | 0.03 |
| | WCVI | 166 | 0.36 | 0.36 | 0.69 | 0.56 | 0.17 | 0.11 | 0.04 |
| | QCS | 549 | 0.27 | 0.28 | 0.60 | 0.64 | 0.16 | 0.18 | 0.06 |
| | SOG | 551 | 0.14 | 0.26 | 0.47 | 0.76 | 0.20 | 0.26 | 0.06 |
| Camera | Coast | 2047 | 0.06 | 0.10 | 0.22 | 0.80 | 0.70 | 0.06 | 0.02 |
| | HG | 818 | 0.10 | 0.15 | 0.34 | 0.79 | 0.30 | 0.26 | 0.10 |
| | NCC | 580 | 0.06 | 0.20 | 0.39 | 0.76 | 0.34 | 0.17 | 0.10 |
| | WCVI | 139 | 0.20 | 0.12 | 0.23 | 0.87 | 0.58 | 0.19 | 0.01 |
| | QCS | 410 | 0.10 | 0.12 | 0.34 | 0.73 | 0.41 | 0.15 | 0.10 |
| | SOG | 196 | 0.21 | 0.11 | 0.23 | 0.86 | 0.58 | 0.12 | 0.07 |
| ROV | Coast | 6059 | 0.15 | 0.24 | 0.42 | 0.83 | 0.30 | 0.23 | 0.04 |
| | HG | 1762 | 0.14 | 0.32 | 0.39 | 0.87 | 0.48 | 0.11 | 0.02 |
| | NCC | 3909 | 0.18 | 0.27 | 0.49 | 0.79 | 0.14 | 0.35 | 0.03 |

model can be attributed to the large, homogeneous Rock predictions of the 100 m model in shallower waters (Fig 4) showing that model performance was dependent on depth and resolution. This confirms our initial belief that the 100 m model would perform better in deeper waters and the 20 m models better in shallow waters. Our qualitative assessment of the mapped predictions agrees with the numerical analysis. Specifically, the 100 m prediction of large contiguous areas of Rock substrate nearshore and the failure of the 20 m model to capture sediments associated with known features (e.g., canyons) in deeper waters (Figs 4 and 8) illustrate how coarser spatial resolutions are unable to represent finer scale heterogeneity in substrate, while the higher resolution needed to capture that heterogeneity can miss larger geomorphic features. This supports the decision to limit the 20 m models to shallower depths, and shows that mapping the different scales of heterogeneity will require multiple resolutions, the integration of which would be best captured using object-based approaches [e.g., 6, 12] where representation is not resolution-dependent.

Our results also support the view that substrate heterogeneity generally decreases with depth, and suggest that resolution-based differences in performance across depths (Figs 6 and 7) are at least in part influenced by the true heterogeneity in different depth classes. For example, the higher accuracy of 20 m models in shallower water is because they can better capture nearshore heterogeneity. Similarly, the more consistent fit of the 100 m model across depth classes in the SOG region can be explained because certain characteristics of the region (a relatively shallow marginal sea dominated by mud in deeper waters [65] and less exposure to wind-wave energy than other regions) combine to minimize differences across depths. This emphasizes the importance of considering process stationarity (see following section). Nevertheless, the correlation of predictive power with depth (Fig 7) suggests that the 20 m models will be more reliable for nearshore studies to about 50 m depth, while the 100 m model would be more reliable in deeper areas.

To assess whether sampling density contributed to model performance we looked for patterns in accuracy to see if it was related to the density of observations (not shown). We found sample density was highly correlated with depth (as expected given the sampling context of much of the build data–see S1 Table in S1 File), making it impossible to disentangle the effect of density from depth with our observations. While understanding the role of sample density

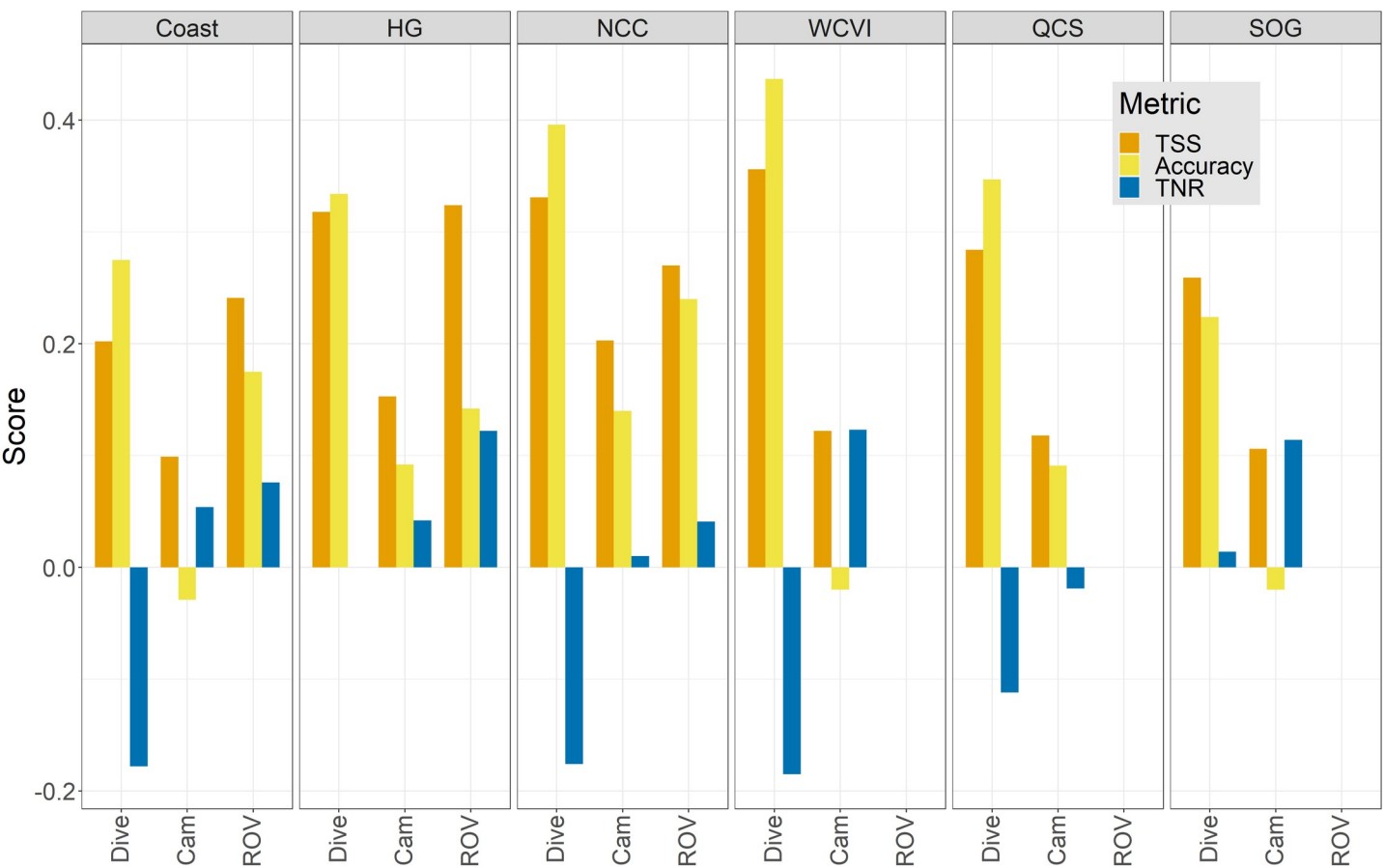

**Fig 9. Aggregated accuracy metrics of predictive power for each independent data set by region.** Accuracies are shown as the difference from the no-information baseline: positive values indicate performance better than random and negative values indicate performance worse than random. The no-information baseline for TSS is always 0.0 because it integrates across classes. However in our error matrices with four classes (S4 Table in S1 File), the baseline for the true positive rate (TPR) is 0.25, and for the true negative rate (TNR) it is 0.75. Missing bars show either a difference of 0 indicating performance no better than random (HG) or missing data (ROV for QCS and SOG).

would provide important information to the design of sampling programs, accuracy is likely to be maximized when sample density is the same or better than the analytic resolution. Ideally, choice of resolution would include an explicit rationale to ensure the resulting product is not misinterpreted by drawing inferences at inappropriate resolutions.

Analytic resolution also influences process stationarity because processes are scale-dependent [66]. This means coarser models will better represent more averaged conditions (which by definition have less variability and correspondingly higher stationary). Thus, our observed differences between the 20 m and 100 m models are also due, in part, to the different processes captured by the different resolutions.

## Process stationarity

Correctly representing driving processes is central to predictive power [67]. However, the reliability of such representations across a seascape depends in part on the assumption of stationarity, a typically tenuous assumption, particularly across larger spatial extents [68]. In this analysis, both class-based results (Fig 2) and variable importance (Fig 5) showed strong evidence for non-stationary processes across regions, while non-stationarity across both regions

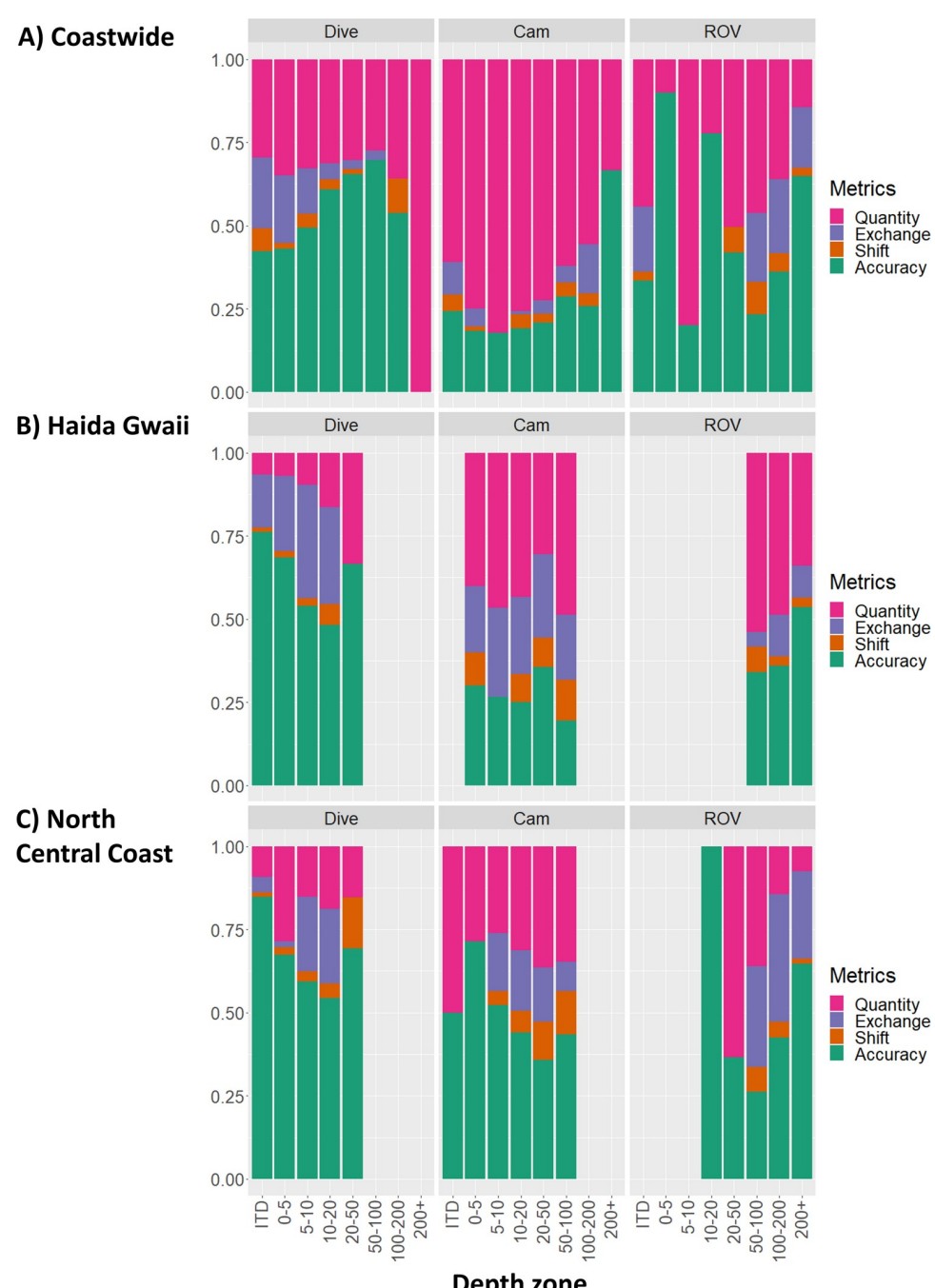

**Fig 10. Error assessment of predictive power by depth zone.** Accuracy and error metrics for the predictive power of the (A) Coastwide, (B) Haida Gwaii, and (C) North Central Coast models for each independent data set (depth zones and regions not shown had insufficient sample sizes).

and depths is suggested by differences in aggregated metrics of model fit (Figs 6 and 7) and error assessment (Fig 9).

There are obvious reasons for non-stationarity across regions. For example, the SOG is strongly influenced by sediments from the Fraser River [65] while exposed regions are more influenced by wind-wave energy. Elsewhere, the unique characterization of variable

importance on the NCC may reflect the competing dominance of tidal energy in channels and wind-wave energy (fetch) in inlets and exposed areas creating within-region differences. Finally, predictions of sand in exposed coastal areas of the WCVI may be made more difficult because the local processes responsible differ from an otherwise strong association between Rock and high fetch. Other (unrepresented) factors such as freshwater input and upland geology will also differ both within and across regions. Given that these processes play out on a crenulated coastline carved up by deep narrow fjords, it would not be surprising if the distribution of substrate types also depended on local processes and geological history.

Such local processes cannot be teased out by classification models. Instead, they are generalized across the model domain according to the prevalence and spatial distribution of the observations. We argue that model reliability is higher in places where predictor values are closer to the center of their ranges, thereby avoiding boundary conditions. This means oceanographic uniqueness will likely be correlated with poor model quality.

## Bias and accuracy

Models of substrate, habitat and climate are regularly built on other models. Thus, any layers used in this way (e.g., bathymetries, current models, remotely sensed primary production) will have their own artefacts, uncertainties and limitations. Careful consideration of data sampling and processing steps such as averaging, aggregation, or interpolation methods is therefore warranted.

In contrast to their consistent validation against the build test data, the predictive power of our models varied across the independent data sets, a clear indication of the differences in data context. Such differences are a function of both data collection and preparation. For example, the occurrence of Dive data deeper than 50 m depth (as implied by the 100 m model, Fig 10A) is an artefact of large pixels in the coastal zone over-generalizing depths. This effect is also evident in the ROV data, which are typically not collected in waters shallower than 20 m. This misallocation of the data is not due to positional inaccuracy of the observations (we screened the depths of the independent data observations for agreement with the 20 m bathymetry) but rather a function of the resolution of the modelled bathymetry. Specifically, since a 100 m raster cell often covers a wide range of actual depths, particularly in steep topography, any spatially associated observation would be assigned to the single raster value regardless of the observation depth. Thus, what can appear to be a positional inaccuracy or bad data can actually be a function of analytic resolution.

This suggests that building models with a compilation of data from different sources can reduce (or average) sampling bias. This also implies that a model built with data from a single sampling context may have lower predictive power. We suggest that using build data compiled from several different sampling contexts will improve model performance because the diversity of biases will force the classifications to be more general, much like the generalization of processes described above regarding stationarity.

Model reliability is also influenced by the number and nature of the classes used in the classification. For example, Rock, Sand, and Mud are all more definitive than our Mixed class which, by definition, included a variety of heterogeneous classes (e.g., sand and cobble, gravelly sand, boulders on silt). With this diversity (i.e., a lack of independence), any method would be hard-pressed to reliably describe such a class. So it's not unexpected Mixed was less well predicted than the more independent classes (see S4 Table in S1 File). However, its consistently high TNR (Fig 2) shows it was rarely predicted in error, perhaps in part due to its relatively small sample size (Fig 3 and S3 Table in S1 File). We suggest such a class is useful to the overall classification because providing a home for less definitive observations reduces the

misclassification of the other classes, and provides insight into local heterogeneity. While an alternative is to have more classes, this can exacerbate the prevalence problem and lead to reduced accuracy [22, 24].

Spatially, we have shown that examining maps of model predictions is also critical to understanding model accuracy. Our qualitative visual assessment identified both poor spatial accuracy in mapped predictions and spatial artefacts, neither of which are apparent in the performance metrics. In addition to non-stationarity and analytic resolution (discussed above), the accuracy of predictors must also be considered. For example, the ocean current model we used (for all models except the 20 m SOG) had a native resolution of 3 km. While the related predictors (tidal and ocean current energy) were the most important predictors in the 100 m model, the interpolation and resampling to 20 m may have served to obscure rather than enhance nearshore variability, despite still helping explain the broader pattern in the observations. Similarly, the artefacts evident in both the 20 and 100 m bathymetric models, themselves the result of sampling bias [e.g., 31], serve as a caution to fitting models too closely to a modelled predictor. These insights suggest we have reached the limits of what can be achieved with our existing predictor data, and that higher resolution predictors that more closely match the analytic resolution are needed to improve model accuracy.

Recognizing artefacts is critical. We suggest all predictor layers be examined carefully before modelling. Our experience shows that artefacts (e.g., for depth) are more clearly visible when examining derivatives. If artefacts are observed, they can often be mitigated by smoothing (S3 Fig in S1 File). Another example relates to the importance of including terrestrial elevations, which can improve bathymetric derivatives (i.e., slope and rugosity) [31] and improved representation of steep-sided rocky shorelines in coastal systems [8]. In the end, we can only develop our models with the best information available. However, it is useful to understand the limits of our predictors, and to explicitly communicate how these limitations should inform model interpretation.

## Measuring performance

Our study shows the value of using both accuracy and error assessment metrics, and of comparing performance across spatial subsets (e.g., regions and depths) and individual classes. The error assessment provided insight into tests of predictive power, showing differences across independent data sets (Table 4) and how errors can be associated with resolution (Table 3). We also found differences in how accuracy metrics respond to class weighting, with the TNR more responsive than Overall Accuracy (Table 3), corroborating the observation by Allouche et al. [17] that prevalence has a greater influence on TNR than Accuracy. Class-based metrics also showed the effect of class weighting on model performance (Fig 2) and contributed to our assessment of spatial non-stationarity (Table 4 and Fig 9) by identifying differences in model performance across regions and depths.

By using well known focal areas our qualitative spatial assessment uncovered important differences between numerical and mapped performance not otherwise apparent. Other, more detailed spatial assessments are possible, but in our case they would be complicated by spatial sampling bias (e.g., the rocky bias of the training data, and the shallow bias of the independent dive data), sample density, and questions of spatial-autocorrelation. Such analyses would therefore be most effective with purpose-collected data.

## Next steps

Our illustration of how depth and resolution influence predictive performance challenges the feasibility of producing a gridded coastwide substrate map at a single resolution, pointing to

the need for an object-based framework to integrate substrate class polygons developed at multiple resolutions. Such efforts could extend existing object-based efforts in the region [6], and take advantage of the work done testing the performance of predictors across resolutions, as has been done in some random forest BS classifications [21].

Independent data collected with dedicated surveys would not only support tests of spatial error, but could be used to test the suggestion that more typical parts of the coast will be better predicted than more unique areas. Such data could also be used to assess sub-regional differences in process. Improving predictor resolution would also help improve classification in shallower waters. For example, the interpolated fetch used in the 20 m regional models could be replaced with a more resolved fetch product. All models would benefit from a coastwide ocean circulation model with a sub-kilometer resolution.

Methodologically, we showed that class weighting affords a similar benefit to a 2-step procedure where the dominant class was modeled separately from the three more balanced ones and then combined [see 69]. Despite the imbalance in our build data, the mapped predictions were more balanced (S4 Fig in S1 File) and more consistent with the known characteristics of these regions [65] and our collective experience with the study area. However, our models did predict the independent Dive data (which, like the build data, were biased towards Rock substrate) with greater accuracy than the more balanced Camera data. This suggests differences in prevalence between build and test data could influence estimates of predictive power. While we found no guidance on whether model accuracy improves with increasing agreement between sample and true prevalence, this highlights the need for further research on the role of imbalance, and how to trade off the accuracy from a balanced training sample against the accurate representation of real-world prevalence. Such studies are critical given the challenges imbalanced data pose to random forest models [19].

Disentangling the effect of sample size from its spatial distribution is also an understudied challenge. Questions include whether model accuracy is higher in areas with higher sampling density, whether higher accuracy is better achieved with a balanced training sample or one that more closely corresponds to real-world prevalence, and whether accuracy is maximized by matching the resolution of the analysis with the sampling resolution. Answers to these questions would improve sampling design and support the development of an object-based approach to integrate features from different resolutions.

Investigating how well models predict more heterogeneous (e.g., Mixed) classes could also help guide model refinement, including the definition of more discrete classes [as proposed, for example, by 6]. To understand the interaction between sampling density and depth, estimates of spatial variability in the training data [e.g., 70], or spatial uncertainty in model predictions [e.g., 33] could also provide insights. Such methods may, however, be more relevant to the ecological models produced using the substrate layer developed here.

Despite the significant challenges facing the classification of BS data collected during bathymetric surveys, meter-scale bathymetry for an increasingly large portion of the coast is becoming available, particularly in deeper waters. These data could be used directly in our analysis at local scales to produce higher resolution outputs and support cross-scale analyses. More detailed, reliable, local MB classifications calibrated to BS data [e.g., 29] would be invaluable as independent data to test the substrate predictions developed here.

## Conclusions

We have produced a set of comprehensive, coastwide maps of marine substrate at resolutions appropriate to nearshore and coastwide analyses (e.g., S5 Fig in S1 File). Compared to the 670,000 km$^2$ classified by Stephens and Diesing [30], our 135,000 km$^2$ study area is about 1/5[th]

the size but is 25 times better resolved (100 m vs 500 m grids). Further enhanced by our 20 m regional models, this contribution is one of the most well-resolved national classifications produced to date. Our spatial assessment shows that our 20 m regional models are suitable for shallower (< 50 m depth), coastal regions while our 100 m model is more suitable for deeper, more homogenous areas of the shelf. Although higher resolution (e.g., meter-scale) models are feasible, they will require higher resolution predictors and will likely have to be limited to regional or sub-regional areas to manage the challenge of stationarity.

We expect model reliability will be highest in more typical, well-sampled areas of the coast, where predictor values were closer to the coastwide or regional average. Predictions in more unique, under-represented areas will be less reliable. Understanding the relationship between resolution and representable features will help users assess the reliability of the mapped predictions.

Our tests of predictive power suggest that building models with data compiled from diverse sampling contexts may improve predictive power by integrating the sampling biases into the models. They also emphasize the importance of distinguishing predictive power from model fit.

Our analysis is one of the few to predict substrate classes from a diverse set of observations over large spatial extents at ecologically relevant scales. Our predictive models are also the first to be evaluated using both accuracy and error metrics illustrating the benefits of comprehensive model assessment. Our maps will contribute to marine spatial planning initiatives in Pacific Canada, and our methods may be useful in other jurisdictions where substrate maps are required.

## Supporting information

**S1 File.**
(DOCX)

## Acknowledgments

Matt Grinnell created the fetch points in Hecate Strait and Queen Charlotte Sound. Input and support from Jessica Finney greatly improved the analysis presented in this paper. We are grateful to CHS for providing their substrate observations, and Peter Wills in particular for insights into characteristics of the CHS data and for making the data from field sheets available. Collegial reviews of an earlier draft by Cooper Stacey and Beatrice Proudfoot, and formal reviews by Gary Greene and David Bowden greatly improved the coherence and readability of this paper.

## Author Contributions

**Conceptualization:** Edward J. Gregr, Dana R. Haggarty, Sarah C. Davies, Joanne Lessard.

**Data curation:** Dana R. Haggarty, Sarah C. Davies, Cole Fields, Joanne Lessard.

**Formal analysis:** Edward J. Gregr, Dana R. Haggarty, Cole Fields.

**Funding acquisition:** Joanne Lessard.

**Investigation:** Edward J. Gregr, Dana R. Haggarty, Sarah C. Davies, Cole Fields, Joanne Lessard.

**Methodology:** Edward J. Gregr, Dana R. Haggarty, Sarah C. Davies, Cole Fields, Joanne Lessard.

**Project administration:** Edward J. Gregr, Joanne Lessard.

**Resources:** Joanne Lessard.

**Software:** Edward J. Gregr, Cole Fields.

**Supervision:** Joanne Lessard.

**Validation:** Edward J. Gregr, Sarah C. Davies, Cole Fields, Joanne Lessard.

**Visualization:** Edward J. Gregr, Dana R. Haggarty, Sarah C. Davies, Cole Fields, Joanne Lessard.

**Writing – original draft:** Edward J. Gregr, Dana R. Haggarty.

**Writing – review & editing:** Edward J. Gregr, Dana R. Haggarty, Sarah C. Davies, Cole Fields, Joanne Lessard.

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
