## [Decision Letter · Decision Letter 0]

6 Apr 2021

PONE-D-20-40989

Comprehensive marine substrate classification of Canada’s Pacific shelf

PLOS ONE

Dear Dr. Gregr,

Thank you for submitting your manuscript to PLOS ONE. After careful consideration, we feel that it has merit but does not fully meet PLOS ONE’s publication criteria as it currently stands. Therefore, we invite you to submit a revised version of the manuscript that addresses the points raised during the review process.

The reviewers find a lack of clarity in many areas, in particular around the maps that may be derived from the model and how they would compare to present maps.  Reviewer 2 offers a way to bring clarity to the presentation of the manuscript that would help readers.

We look forward to receiving your revised manuscript.

Kind regards,

Judi Hewitt

Academic Editor

PLOS ONE

Journal Requirements:

[The authors declare no competing interests.].   

We note that one or more of the authors are employed by a commercial company: SciTech Environmental Consulting

4. We note that Figures 1, 4, 8, S3 and Striking Image in your submission contain map (Fig. 1) / satellite (Fig. 4, 8, S3 and Striking Image) images which may be copyrighted. All PLOS content is published under the Creative Commons Attribution License (CC BY 4.0), which means that the manuscript, images, and Supporting Information files will be freely available online, and any third party is permitted to access, download, copy, distribute, and use these materials in any way, even commercially, with proper attribution. For these reasons, we cannot publish previously copyrighted maps or satellite images created using proprietary data, such as Google software (Google Maps, Street View, and Earth). For more information, see our copyright guidelines: http://journals.plos.org/plosone/s/licenses-and-copyright.

You may seek permission from the original copyright holder of Figures 1, 4, 8, S3 and Striking Image to publish the content specifically under the CC BY 4.0 license. 

If you are unable to obtain permission from the original copyright holder to publish these figures under the CC BY 4.0 license or if the copyright holder’s requirements are incompatible with the CC BY 4.0 license, please either i) remove the figure or ii) supply a replacement figure that complies with the CC BY 4.0 license. Please check copyright information on all replacement figures and update the figure caption with source information. If applicable, please specify in the figure caption text when a figure is similar but not identical to the original image and is therefore for illustrative purposes only.

Reviewers' comments:

Reviewer's Responses to Questions

**Comments to the Author**

1. Is the manuscript technically sound, and do the data support the conclusions?

Reviewer #1: Partly

Reviewer #2: Partly

2. Has the statistical analysis been performed appropriately and rigorously? 

Reviewer #1: I Don't Know

Reviewer #2: Yes

3. Have the authors made all data underlying the findings in their manuscript fully available?

Reviewer #1: Yes

Reviewer #2: No

4. Is the manuscript presented in an intelligible fashion and written in standard English?

Reviewer #1: Yes

Reviewer #2: Yes

5. Review Comments to the Author

Reviewer #1: This is a fairly well written manuscript that describes the construction and use of a statistical model based on random forest classification using bottom type data to map substrate along Canada’s Pacific margin. While the authors extensively describe their modelling development and provide a convincing argument for the model’s validity, I found that a good comparison of the model with detailed marine benthic habitat maps that exist for the region was missing. Unfortunately, I am not qualified to fully evaluate the statistical approaches described in the manuscript, so focused on the practical aspects of constructing substrate maps.

While I think that the model could be very helpful in making a first approximation of substrate distribution when good multibeam bathymetric data is not available, but sediment samples and other seafloor data is at hand, I am concerned that it could also be misleading. Generally, I found the modeling concept sound and the authors do a good job at testing the model’s applicability. I will leave it to those better qualified than me to evaluate the statistic.

I have a few comments to make about the text. While the text is generally clear there are some areas that I found in need of further explanation or clarification. I have added these few comments in the pdf. However, a few points need to be addressed here. While the manuscript is well referenced, and the references cited appear appropriate and comprehensive, the citation format is mixed (i.e., numbered as per the journal’s format in places while in other places not numbered). In addition, I have provided below some references that the authors may find useful, especially in regard to their discussion of “mixed” substrate types and that might be useful in validating the model.

The term “mixed substrate” needs to be better explained and defined when first used. The authors use backscatter as data to identify substrate types but do not fully explain how some “soft” substrate clasts such as gravel, cobble and pebbles can form a hard substrate type as a gravel-pebble-cobble pavement, which is quite common in the offshore BC region.

One confusing problem appears in the comparison of the 20 m and 100 m resolution models where a submarine canyon is not identified in the 20 m model. I am not sure why this is the case. If a distinct change in depth occurs with the presence of a canyon, why would the model not detect this? Is it because the canyon is too large? If so, what about gullies and small features, would they also not be identified in the 20 m model?

My recommendation is that the manuscript be published after minor modifications and clarification of points raised in this review are addressed. I would be especially interested in further validation of the model using available published Canadian marine benthic habitat maps for the Southern Georgia Strait region (see Greene and Barrie, editors (2011). This would show how well the authors’ model fits with comprehensive habitat maps based on MBES data interpretations.

Suggested pertinent references:

Greene, H.G., Yoklavich, M.M., Starr, R., O’Connell, V.M., Wakefield, W.W., Sullivan, D.L. MacRea, J.E. and Cailliet, G.M., 1999. A classification scheme for deep-water seafloor habitats: Oceanographica ACTA, v. 22, n. 6, p. 663-678.

Greene, H.G., Yoklavich, M.M., O’Connell, V.M., Starr, R.M., Wakefield, W.W. and Cailliet, G.M., 2000, Mapping and classification of deep seafloor habitats: ICES paper CM 2000/T:08, 11 p.

Greene, H.G., Bizzarro, J.J., Tilden, J.E., Lopez, H.L., and Erdey, M.D., 2005. The benefits and pitfalls of geographic information systems in marine benthic habitat mapping: In Wright, D.J. and Scholz, A.J., (Eds.), Place Matters. Oregon State University Press, Portland, OR, 34-46.

Greene, H.G., Bizzarro, J.J., O’Connell, V.M., and Brylinsky, C.K., 2007. Construction of digital potential marine benthic habitat maps using a coded classification scheme and its application: In Todd, B.J., and Greene, H.G. (Eds.), Mapping the Seafloor for Habitat Characterization, Canadian Geological Association Special Paper 47, 141-155.

Reviewer #2: PONE-D-20-40989 review

Overall comments

This manuscript describes an interesting and potentially very useful initiative that represents a lot of work and will make a good paper. At present, however, I find its presentation is unclear, with too much blurring between the conventional sections of introduction, methods, results, and discussion. I also have some questions about how the analyses were performed and the inferences made from the results.

The aim of this study is to generate reliable maps of seabed substrate type for the Pacific coast and continental shelf of Canada. The authors compile ca. 200,000 point records of seabed substrate type and use a tree-based machine-learning method, Random Forest (RF), to develop correlations between the observations and gridded layers for depth, depth-derivatives, and seabed wave energy. These correlations are then used to predict substrate type as a continuous layer across the region. The authors then evaluate these predictions by both cross-validation and against a set of independent seabed observations.

The main components of interest in the study, therefore, are: (1) the provenance and spatial distribution of the point observations; (2) the provenance and accuracy of the predictor variables; (3) the provenance and spatial distribution of the independent test data; (4) the modelling methods used, and (5) the resulting maps, with the final mapped classifications being the most important.

What we actually get in the ms is a lot of detail and talk about how well the modelling method has been applied but not much detail, or clarity, on what, to me, are the main points of interest (1), (2), (3), and (5). I found the sequence of sections and content confusing and difficult to follow, with elements of introduction, methods, results, and discussion jumbled up together throughout. As a simple example, why not start the Methods with a description of the study area instead of a statement about the modelling method used and how good it is? The important, and useful thing about the study, by my reading at least, is that it brings together all available point sample data for the region and uses them to generate a classification of the entire Pacific continental shelf of Canada. The methods are important but there is already plenty of published information available about how different modelling methods and evaluation metrics compare, and at present in this ms over-emphasis of these details eclipses appreciation of the main achievement of the study: the compilation of the input data set and the new maps of substrate type generated from them.

The Introduction stretches to 8 pages, includes much material that would be more appropriate in the Discussion, and does not, to me at least, seem to follow a logical progression. The Introduction should provide a concise background to the study; why and where it was undertaken, background and issues associated with the area of research, methods available, and what the specific objectives of this study.

The statements given at the start of the Results section are an example of one of my main issues with the way the study has been presented: it places all the emphasis on the modelling methods and very little on the input data compilation, particularly in terms of spatial distribution, the credibility of the predictor variables, or, most importantly, the utility of the final outputs. It reads more as a methods paper than an attempt to produce a useful resource for environmental management (which, of course, is what it actually is).

Specific comments

Abstract

The abstract is a concise summary of the study; summarising background, aim, study area, data and methods used, results, and conclusions. If the body of the manuscript followed this simple, logical, and interpretable structure, this would be a very nice paper. As it stands, I find the subsequent sections muddled, over-long, and confusing.

Line 28-30: “Predictive power was lower … when models were evaluated with independent data sets, emphasising how this is different from model fit”. This kind of statement seems a bit disingenuous to me, suggesting that ‘model fit’ by cross-validation using subsets of the training data and performance against fully independent survey data are of equal value in assessing the utility of a model. The real test of any model, particularly those designed to inform environmental management decisions, is how well its predictions match reality, in the form of independent observations. Perhaps no need to include “emphasizing how this is different from model fit” (it tells us nothing after all) but add more explanation in the discussion about how the models performed against independent data.

Lines 37-38: “This understanding relies on models of habitat suitability …”. This seems a sweeping statement to me when “This’ covers all aspects of marine ecosystem management.

Lines 38-39: “The credibility of which depends in large part on the accuracy of the underlying environmental predictors.” Yes, this is very true but I would observe that the final layers you develop here are based on the same techniques (RF) and thus carry the same issues of uncertainty associated with the predictor variables.

Line 46: need to reference Random Forest at first use.

Line 51: “mobilizing available observations …”: how do you ‘mobilize’ observations?

Lines 56-57: Meaning of this sentence is unclear to me: at this stage, the reader has no idea what is meant, in this context, by “weighting for prevalence”, and presumably the meaning is ‘use of diverse evaluation metrics’, and what does ‘qualitative assessment’ refer to here?

Line 61: Prior to MBES, seabed characteristics were derived from empirical point observations, which were often accurate and, for older lead-line records, included physical samples of the seabed. The 'inference' element comes when continuous bathymetry layers are created. For most of the world's oceans and seas, this is still the case.

Line 69: Need to be more concise with language: as written here, the meaning is “comprehensive surveys are particularly expensive and time consuming for less developed countries …”. The time and expense are the same, whatever the economic status of the country, its just the affordability that differs. And the final clause of the sentence doesn’t match its subject (i.e. “comprehensive acoustic surveys … can take decades to completely map.”

Line 74: How much, approximately of the shelf here has been mapped?

##: “… a diversity of metrics …”. Has no meaning; just tell us what you used.

Methods

Lines 221-223: should be in the Introduction.

Line 226: “nested” needs to be defined, i.e., nested within what? I assume within the ‘coastwide’ model but this is not explicit in your sentence.

Line 227: “paired models”, meaning what? Without clear explanation these terms are meaningless to the reader.

Lines 219-240: These first three paragraphs of the Methods are also a prime example of what I struggle with in the presentation of this study. They give a condensed summary, an abstract in effect, of what the study did but without any detail. This level of explanation would work well in the Introduction but here, in the Methods, it’s just confusing. For instance, the most fundamental aspect of the study is the input dataset of substrate type observations: this is the first thing the reader needs to be told about, in detail, to be able to understand what the subsequent models are working on and thus assess whether the resulting maps make sense. At present, the input data appear almost as an afterthought, with just a passing reference to Table 1 and no explanation of the data provenance, spatial distribution, or reliability.

Lines 259-260: But how was the weighting done: more weight to higher prevalence? Need explicit methods descriptions.

Lines 270-272: There is not enough detail on how this partition into training and test partitions was done. For spatial data, the way in which test data are selected can have strong influence on subsequent evaluation metrics. Were the test data selected at random, or in spatial bands, or by a more sophisticated spatially disaggregated method? Also, the wording here and later implies that only one iteration of each model was generated, all using the same partition of training and test data. If that is what was done, explanation is needed as to why k-fold cross validation (multiple iterations of each model, each iteration using a different split of the input data between training and test) was not conducted.

Line 275 and onwards: “Addressing our objectives”. Why is this a subsection in the Methods? Too much of the text here should really be in the Introduction or Discussion, not here in the Methods.

Line 280: “We weighted classes according to their prevalence”. Again, how? Weighted up, or down, and by what proportion?

Line 285-286: The input variables of this study, both response and predictor, are relatively very simple, representing primarily (entirely?) physical factors. I am not convinced by the argument that the physical process factors here should be expected to be non-stationary. I suspect differences in the density (‘prevalence’) and reliability of input response and predictor data will have a more important influence on outcomes than non-stationarity of processes.

Line 306 and onwards: “Model evaluation”. Again, by my reading of it, far too much wordage that should be (or already is) covered in the introduction or discussion. I might be jaded but much of this reads like rehashed material from textbooks. The point is, however, that I am used to working with these kinds of data and this kind of modelling method, and the further I read here, the more I find myself confused as to what was done and why.

Line 347: “Model build data”. At last! But there is no detail given about the spatial distribution of these data. For interpretation of the results, I would argue that it is essential to show the reader maps showing how these input data are distributed in space.

Line 379 and onwards: “Independent evaluation data”. How did you decide on which data to include in the ‘build’ set and which in the ‘independent’ set? Both sets include DFO Dive and ROV data, so how do these differ from the cross-validation test data withheld from the training dataset? If the two set of data are actually just arbitrary subsets from the same sources, the independence of the ‘independent’ dataset would be questionable. Again, needs clearer explanation of basic details.

Results

As with the methods, I find the sequence of sections here to be unintuitive, and the content to mix results with discussion material.

Lines 404-407: This paragraph is discussion material.

Line 533: Ah ha. Here, at last, we have more detail about the independent data but still, I would say, not enough to assess their utility. For instance, N = nearly 2,500 ROV ‘mud’ observations for the coastwide model domain but if each observation represents records of substrate type at 20 m intervals along seabed transects each one of which might be one or more km long (at 50 records per km), these data are likely to be strongly clumped in space. If you have not taken measures to account for this spatial clumping of the data, the resulting metrics of performance are likely to be unreliable and probably inflated. We need to see how these records are distributed in space to be able to assess whether the results are useful or not.

Results in general: I would have found if much more useful and interpretable to have included both cross-validation and independent test scores in the same table, simplified down to just one or two example metrics: all the rest could go into the Supplementary Material. Also, a question: where can the final map outputs be found? If the aim was to generate mapped predictions for use in environmental management, the outputs need to be accessible.

Discussion

I found this section to read better than the others. I have not made detailed notes but I would make the same observation about the inferences around stationarity: given the imbalances in the spatial distribution and provenance of your sample and test data, can you really be sure that the differences in model performance you see among regions is attributable to non-stationarity in environmental process rather than artefacts in your input data?

6. PLOS authors have the option to publish the peer review history of their article (what does this mean?). If published, this will include your full peer review and any attached files.

Reviewer #1: **Yes: **H. Gary Greene

Reviewer #2: **Yes: **David Bowden

---

## [Author Response · Author response to Decision Letter 0]

1 Jul 2021

Comprehensive marine substrate classification applied to Canada’s Pacific shelf

Manuscript # PONE-D-20-40989

Response to reviewers

2021/06/27

Summary

The manuscript has been extensively re-organized largely in response to Reviewer #2’s concerns about clarity of presentation. The details of this re-organization are described in specific responses below. In addition, we have reviewed the entire paper for unnecessary jargon and clarity, removed redundancies, and added definitions where required. For example, we found 25 occurrences of the word ‘context’, which we have now restricted to the meaning we defined (sampling context). We replaced the remaining occurrences with more precise terminology. 

All comments from Reviewer #1 were addressed and are described in the Specific comments section below. Reviewer #2 provided more general comments, which motivated more significant changes to the paper. In the following sections, we have italicized all reviewer comments and indented our response below them. 

General comments (Reviewer #2)

“the presentation is unclear, with too much blurring between the conventional sections of introduction, methods, results, and discussion”.

“I found the sequence of sections and content confusing and difficult to follow, with elements of introduction, methods, results, and discussion jumbled up together throughout. As a simple example, why not start the Methods with a description of the study area instead of a statement about the modelling method used and how good it is? “

“The Introduction stretches to 8 pages, includes much material that would be more appropriate in the Discussion, and does not, to me at least, seem to follow a logical progression. The Introduction should provide a concise background to the study; why and where it was undertaken, background and issues associated with the area of research, methods available, and what the specific objectives of this study.”

We improved the clarity of the paper by re-organizing the text into a more standard format. We moved discussion text from the Results to the Discussion, and swept up other scattered introductory material into the Introduction. While the Introduction still runs to 8 pages, it has a more logical flow and contains more relevant background information. This re-organization also removed much of the redundancy in the paper. 

We have re-written the challenges section to make them more concise, and to directly address several reviewer comments. We have more clearly integrated the classification work by Greene and colleagues in the section on Ecological relevance. We have replaced the Methods section ‘Addressing the challenges’ with ‘Model development and comparisons’. 

“The main components of interest in the study are: (1) the provenance and spatial distribution of the point observations; (2) the provenance and accuracy of the predictor variables; (3) the provenance and spatial distribution of the independent test data; (4) the modelling methods used, and (5) the resulting maps, with the final mapped classifications being the most important. … What we actually get in the ms is a lot of detail … about how well the modelling method has been applied but not much detail or clarity on … the main points of interest (1), (2), (3), and (5).” 

“The methods are important but there is already plenty of published information available about how different modelling methods and evaluation metrics compare, and at present in this ms over-emphasis of these details eclipses appreciation of the main achievement of the study: the compilation of the input data set and the new maps of substrate type generated.”

“The statements given at the start of the Results section are an example of one of my main issues with the way the study has been presented: it places all the emphasis on the modelling methods and very little on the input data compilation, particularly in terms of spatial distribution, the credibility of the predictor variables, or, most importantly, the utility of the final outputs. It reads more as a methods paper than an attempt to produce a useful resource for environmental management (which, of course, is what it actually is).”

We are very pleased the reviewer sees value in our resulting maps. However, we believe our work has broader utility than to just inform the substrate classification of the Canadian Pacific Shelf and therefore worked hard to fully describe the methods we applied (and to provide code) to allow the methods to be applied in other regions faced with the same challenges and objectives. We therefore see the paper as an application of novel methods to a particular case study, and have reflected this in the revised title “Comprehensive marine substrate classification applied to Canada’s Pacific shelf.”

We included new introductory paragraphs where we now more clearly state our objectives, and have elevated the descriptions of all the data sources to the front of the Methods section. We also added a figure to the supplemental materials to more clearly describe the spatial distribution of the substrate observations. 

We agree that with the reviewer that “there is already plenty of published information available about how different modelling methods and evaluation metrics compare”, but point out that we are 1) presenting a novel application of random forest modelling, and 2) advancing the important step of model validation through an assessment of methodological challenges using a novel assemblage of metrics curated from three disciplines. We think that others faced with a need for broad scale substrate maps will find value in the methods developed here to classify data with coarser resolution and wider geographic extents (compared to the methods more commonly applied to small extent, high-resolution data).

- We have de-emphasized the accuracy of the predictor variables in the introduction. 

“I also have some questions about how the analyses were performed and the inferences made from the results.”

These are elaborated on the Specific comments, below. 

Specific Comments

Reviewer #1: Comments received via marked-up PDF. 

[1] line 48: All references checked for citation style. Thank you. 

[2] line 48: “What is mixed? Does this mean soft and hard like defined by Greene et al. (1999; 2007) where they use induration (soft, hard, mixed hard & soft) to describe mixed? I see you define mixed later on in the manuscript but it should be defined when first used. Also, see Greene & Barrie”.

We have added a sentence emphasizing that our classes our ecologically rather than geologically derived. The sentence also refers the reader to both the descriptions of the source data in Table S1 and to the reference with the ecological rationale for the classes. 

[3] line 71: Suggested edit made.

[4] lines 111/112: “Perhaps in the EUNIS system this is correct but BS goes further than just grain size prediction and can be interpreted to apply to packing, sorting, and density. Other uses of backscatter are for mapping facies changes in soft sediment as well as fractured and deformed bedrock in hard substrates.”

We have re-phrased the first sentence to make it clear we were referring to the applications of backscatter classification (BS) to the EUNIS application. 

[5] lines 113/114: In response to our assertion that classifications that integrate hand and soft substrates are rare in the literature, the reviewer stated “classifications that discuss the relationship of hard and soft substrates in a mixed substrate category has been discussed and defined in Greene et al. (1999; 2007).”

We appreciate the reminder of existing comprehensive classifications and have included the appropriate refence in this section. 

[6] line 120: The definition of Mud has now been included earlier in the paper (see line 48 comment above) and includes ‘unconsolidated’ as a qualifier.

[7] line 150: Edit made. Thank you.

[8] line 165: Reviewer comment: “Why restrict elevation variations to landslides, why not to moraines and other irregular morphologic features that have distinct elevation changes? Also, not all landslides have distinct elevation changes, for example debris flows, or mudflows are very subtle topographic features. Sediment sampling by itself is not the best way to map marine benthic habitats, other process interpretations need to be applied.”

We thank the reviewer for their thoughts on morphological features. For clarity, and to avoid confusion with the word “landslide”, we have replaced the term “land-side” with “terrestrial” throughout the manuscript. “Landslides” are not part of this analysis. 

[9] lines 248 to 250: Reference formats are inconsistent. 

We thank the reviewer for noting the incorrect format and absence of the tabled references. This has been corrected. 

[10] line 262: “What does this mean? Rocky reef is not a geologic term but more of a maritime term and as such is confusing. I like to think of a reef being biological and a shoal or bank composed of rock being a habitat feature comprised of rock and not calcium carbonate or silica.”

We appreciate the semantic complexities that emerge when synthesizing work from different disciplines and appreciate the reviewer sharing their perspective. 

We have revised the sentence for clarity by removing the reference to rocky reefs. 

[11] line 359: The reference has been corrected. Thank you. 

[12] line 376: The reference has been corrected. Thank you. 

[13] line 419: “This seems reasonable and not surprising as rock and mud are generally pretty stable substrates if consolidated.”

This comment reflects the effectiveness of the methods for these two classes. No change. 

[14] line 601: “This term appears out of place as in geology structure relates to folds and faults and other such structures and in biology it refers to organisms. I suspect what is meant here is bottom geomorphology as that is what the scale or size of a feature is that is being identified.”

We thank the reviewer for pointing out the potential confusion of the phrase ‘bottom structure’. We have changed this to ‘substrate’ to clarify our meaning, and have added the qualifier ‘geomorphic’ at the end of the sentence to clearly express our meaning. These clarifications enabled further improvements to the text in the subsequent sentence. 

[15] line 606: “I would encourage some caution here as substrate heterogeneity may become more uniform with depth along some continental slopes but in areas that are heavily gullied and incised by submarine canyons this is not always the case. Increased substrate heterogeneity is especially prevalent along tectonically active margins such as the Queen Charlotte transform margin of Canada.”

We appreciate the reviewer’s insight into the regional effects of plate tectonics. Our assertion here is a reflection of the available substrate data and the analysis. The inclusion of the qualifier ‘generally’ was intended to acknowledge that the observed trend may not be the case everywhere. No change. 

[16] lines 610/613: “I see where the authors are going with this but do not agree that the coastline in the Strait of Georgia is more homogeneous than the outer coast, especially since the region has been tectonically deformed and more so altered by glaciation.”

We have removed our observational claim regarding the homogeneity of the coastline, and emphasized the energetic differences between this and the other regions. 

[17] line 631: The reference has been corrected. Thank you. 

[18] line 636: The reference has been added. Thank you. 

[19] line 638: What happened to Reference #63, I did not see it cited before this reference. 

Reference #63 is cited above on line 612. 

[20] line 644: “What about tidal exchanges. I would suspect that the boundary conditions of the SOG would cause localized scouring and deposition even though Fraser River sedimentation is active.”

We agree that localized scouring is likely in areas of high energy and mention the importance of such local processes later in the paragraph. However, this does not invalidate the acknowledged role of sedimentation in the SOG. We have added a reference to support this assertion, and re-phrased the start of the paragraph to make our point more clearly.

[21] line 652: ”I would expect that geologic history plays a role as well.” 

We have rephrased this sentence to refer to “local processes and geological history”. 

[22] line 675/677: “Mixed class should be defined up front when you first discuss it.”

Addressed as part of reviewer’s comment on line 48 (above). 

[23] line 683: “Not sure what this means. Do you refer to expert interpretations of say MBES data or something else. This appears to be a critical point as this would allow some validation of your model.”

We have revised the phrase ‘mapped predictions’ to ‘maps of model predictions’ to improve the clarity of this statement. 

[24] line 700: The confusion between land-side and landslide has been addressed as part of the reviewer’s comment on line 165. 

[25] line 703/704: “Such as what type of terrestrial data? Are you referring to LiDAR, satellite imagery, or what?”

We have changed ‘terrestrial data’ to ‘terrestrial elevations’ to more clearly make this point, and largely re-written the middle part of this paragraph for clarity. 

[26] line 709: “Seems to be out-of-order.”

This comment refers to reference [63] appearing after [51]. We note that at this point in the manuscript, citations may not appear in order as many were cited earlier. 

[27] line 740/742: “It would seem to me that applying you model to the habitat maps that were published by Greene and Barrie (2011) would be helpful in validating you work. The MBES data, sediment samples in the form of grabs and cores, bottom photos and other data used in the construction of their marine benthic habitat maps are readily available and could be inputted into your model.”

We considered doing this validation, however we chose to not complicate an already lengthy analysis because the Greene and Barrie (2011) analysis covers a very small portion of our study area, and because earlier work (Gregr et al. 2013) used a more simplistic sediment classification and showed reasonable agreement. No change. 

[28] line 756: “Not sure what exactly "boundary class" means here.”

We thank the reviewer for noticing this introduced jargon. We have removed the phrase “boundary class” and use the more explicit ‘heterogeneous class’ along with an example. We have also made additional clarifications regarding the relevant next step in the rest of the paragraph. 

[29] line 766/767: “From multibeam echosounder data distinct geomorphology can be machined ID and refined by expert knowledge. This should be an approach that is useful to you.”

We are aware of this method and describe it in the introduction where we also note that a dependence on multibeam echosounder data is a hinderance for comprehensive classification for many jurisdictions. This lack of comprehensive multibeam coverage was a key motivation for our work. No change. 

Reviewer #2: PONE-D-20-40989 review

Abstract

The abstract is a concise summary of the study; summarizing background, aim, study area, data and methods used, results, and conclusions. If the body of the manuscript followed this simple, logical, and interpretable structure, this would be a very nice paper. As it stands, I find the subsequent sections muddled, over-long, and confusing.

We have revised the manuscript as suggested by the reviewer as described in the responses to the following comments. 

[30] line 28-30: “Predictive power was lower … when models were evaluated with independent data sets, emphasizing how this is different from model fit”. This kind of statement seems a bit disingenuous to me, suggesting that ‘model fit’ by cross-validation using subsets of the training data and performance against fully independent survey data are of equal value in assessing the utility of a model. The real test of any model, particularly those designed to inform environmental management decisions, is how well its predictions match reality, in the form of independent observations. Perhaps no need to include “emphasizing how this is different from model fit” (it tells us nothing after all) but add more explanation in the discussion about how the models performed against independent data.

We have removed the sentence fragment in question from the abstract, and have revised the text in response to various comments below to emphasize the difference between model fit and model predictive power, as well as clarifying the methods applied. 

[31] lines 37-38: “This understanding relies on models of habitat suitability …”. This seems a sweeping statement to me when “This’ covers all aspects of marine ecosystem management.

We have re-written the introduction for clarity and removed this phrase.

[32] lines 38-39: “The credibility of which depends in large part on the accuracy of the underlying environmental predictors.” Yes, this is very true but I would observe that the final layers you develop here are based on the same techniques (RF) and thus carry the same issues of uncertainty associated with the predictor variables.

Our revised introduction now clearly describes how our work fits within the constellation of models we believe are necessary to support coastal resource management. 

[33] line 46: need to reference Random Forest at first use.

Done by moving line up from first paragraph of the discussion.

[34] line 51: “mobilizing available observations …”: how do you ‘mobilize’ observations?

We have replaced ‘mobilize’ with the phrase ‘making use of’. 

[35] lines 56-57: Meaning of this sentence is unclear to me: at this stage, the reader has no idea what is meant, in this context, by “weighting for prevalence”, and presumably the meaning is ‘use of diverse evaluation metrics’, and what does ‘qualitative assessment’ refer to here?

We have removed the closing paragraph of the introduction containing this sentence, which was intended to foreshadow some of the more technical results. 

[36] line 61: Prior to MBES, seabed characteristics were derived from empirical point observations, which were often accurate and, for older lead-line records, included physical samples of the seabed. The 'inference' element comes when continuous bathymetry layers are created. For most of the world's oceans and seas, this is still the case.

We have replaced to phrase ‘inferred from’ with ‘based on’.

[37] line 69: Need to be more concise with language: as written here, the meaning is “comprehensive surveys are particularly expensive and time consuming for less developed countries …”. The time and expense are the same, whatever the economic status of the country, its just the affordability that differs. And the final clause of the sentence doesn’t match its subject (i.e. “comprehensive acoustic surveys … can take decades to completely map.”

We have revised the relevant sentences, and made some additional edits to this section for clarity. Thank you. 

[38] line 74: How much, approximately of the shelf here has been mapped?

We have added an estimate of the amount of multi-beam completed.

[39]: “… a diversity of metrics …”. Has no meaning; just tell us what you used.

We could not find this phrase in the manuscript. However, we do refer to ‘diverse and interpretable metrics’ at the end of the first Introduction section. We disagree with the reviewer that this has no meaning. We believe this sentence has value in foreshadowing the importance of metrics selection, and believe this preferable to listing the many metrics used and the rationale for their selection. 

However, the re-organization of the manuscript prompted by various other comments by this reviewer have made the relevant section in the Methods more apparent and accessible. 

[40] lines 221-223: should be in the Introduction.

We agree and have moved the sentence.

[41] line 226: “nested” needs to be defined, i.e., nested within what? I assume within the ‘coastwide’ model but this is not explicit in your sentence.

We removed the word ‘nested’ and revised the sentence for clarity. 

[42] line 227: “paired models”, meaning what? Without clear explanation these terms are meaningless to the reader.

We have re-phrased for clarity, replacing “paired” with the more explicit “with and without class weights”.

[43] lines 219-240: These first three paragraphs of the Methods are also a prime example of what I struggle with in the presentation of this study. They give a condensed summary, an abstract in effect, of what the study did but without any detail. This level of explanation would work well in the Introduction but here, in the Methods, it’s just confusing. For instance, the most fundamental aspect of the study is the input dataset of substrate type observations: this is the first thing the reader needs to be told about, in detail, to be able to understand what the subsequent models are working on and thus assess whether the resulting maps make sense. At present, the input data appear almost as an afterthought, with just a passing reference to Table 1 and no explanation of the data provenance, spatial distribution, or reliability.

The intent of the preamble in the methods section is to give the reader an overview of what is to come. The overall analysis has many aspects and we believe this overview provides considerable value as a guide the reader. ‘

However, in response to the reviewer’s comments, we have limited this to one paragraph, which also now references the data overviews (Table 1 and Table 2) in the first sentence. 

We follow this paragraph with the three sections that detail each of the source data sets (Model build data, Predictor data, and Independent evaluation data). Each of these sections refers to associated tables in the supplemental materials containing the details about the observations the reviewer seeks: Table S1 describes the observations and their preparation for this analysis in some detail. While Table S2 describes the source of the predictor data. We welcome the reviewers comments on the sufficiency of these tables.

We thank the reviewer for prompting this valuable re-organization.

[44] lines 259-260: But how was the weighting done: more weight to higher prevalence? Need explicit methods descriptions.

We have simplified the paragraph to focus on the need for weighting and explicitly described how the class weights were calculated.

[45] lines 270-272: There is not enough detail on how this partition into training and test partitions was done. For spatial data, the way in which test data are selected can have strong influence on subsequent evaluation metrics. Were the test data selected at random, or in spatial bands, or by a more sophisticated spatially disaggregated method? Also, the wording here and later implies that only one iteration of each model was generated, all using the same partition of training and test data. If that is what was done, explanation is needed as to why k-fold cross validation (multiple iterations of each model, each iteration using a different split of the input data between training and test) was not conducted.

We have reorganized this section so that the data partitioning is more clearly described, as is the use of the independent data. The reviewer’s questions regarding model validation are now addressed immediately after this paragraph, in the model evaluation section (see response to comment [49] below. 

[46] line 275 and onwards: “Addressing our objectives”. Why is this a subsection in the Methods? Too much of the text here should really be in the Introduction or Discussion, not here in the Methods.

We agree with the reviewers assessment and have moved much of this material to the Introduction, where it now provides additional background on each of our stated objectives. This allowed us to consolidate several redundant paragraphs. We have renamed this much shortened section to ‘Model development and comparisons’. 

[47] line 280: “We weighted classes according to their prevalence”. Again, how? Weighted up, or down, and by what proportion?

We now explicitly describe class weighting in the new Model development section. 

[48] line 285-286: The input variables of this study, both response and predictor, are relatively very simple, representing primarily (entirely?) physical factors. I am not convinced by the argument that the physical process factors here should be expected to be non-stationary. I suspect differences in the density (‘prevalence’) and reliability of input response and predictor data will have a more important influence on outcomes than non-stationarity of processes.

It’s unclear to us why the nature of the predictor and response variables (simple and physical) is relevant to the question of process stationarity. Nevertheless, we removed this paragraph as part of our response to reviewer’s comment [46] above. 

We also address the part of this comment regarding the density of points with our response to comment [54] below, with the addition of a new supplemental figure showing the random distribution of the build data. And while the predictor data sets are likely to contain artefacts as we discuss, there is no reason to think these could be responsible for the results we found in terms of regional responses, which we draw on for our conclusions about model stationarity. 

Further information is provided in response to comment [57] below. No changes. 

[49] line 306 and onwards: “Model evaluation”. Again, by my reading of it, far too much wordage that should be (or already is) covered in the introduction or discussion. I might be jaded but much of this reads like rehashed material from textbooks. The point is, however, that I am used to working with these kinds of data and this kind of modelling method, and the further I read here, the more I find myself confused as to what was done and why.

This section is central to the comprehensive assessment of model performance and as such forms a critical part of this contribution. While we understand and appreciate the reviewer’s perspective that this information is not novel, we believe we have synthesized this important issue in a way that makes it more accessible to less experienced practitioners.

However, we have carefully reviewed and re-written the section for clarity, removing redundancies and moving some text up to the introduction. 

[50] line 347: “Model build data”. At last! But there is no detail given about the spatial distribution of these data. For interpretation of the results, I would argue that it is essential to show the reader maps showing how these input data are distributed in space.

We have moved the section on build data near the top of the methods section, and added a supplemental figure showing the spatial distribution of the sampling data. 

[51] line 379 and onwards: “Independent evaluation data”. How did you decide on which data to include in the ‘build’ set and which in the ‘independent’ set? Both sets include DFO Dive and ROV data, so how do these differ from the cross-validation test data withheld from the training dataset? If the two set of data are actually just arbitrary subsets from the same sources, the independence of the ‘independent’ dataset would be questionable. Again, needs clearer explanation of basic details.

We have clarified that while the build and independent evaluation data were collected using similar methods, they were collected at different times, often by different observers, and for a different purpose. 

[52] As with the methods, I find the sequence of sections here [Results] to be unintuitive, and the content to mix results with discussion material.

We reviewed the Results section and moved all discussion extending beyond one sentence to the Discussion section. Individual sentences interpreting the results were retained to allow us to convey the reasons for the results more clearly. 

53] lines 404-407: This paragraph is discussion material.

We have moved the paragraph to the beginning of the discussion section.

[54] line 533: Ah ha. Here, at last, we have more detail about the independent data but still, I would say, not enough to assess their utility. For instance, N = nearly 2,500 ROV ‘mud’ observations for the coastwide model domain but if each observation represents records of substrate type at 20 m intervals along seabed transects each one of which might be one or more km long (at 50 records per km), these data are likely to be strongly clumped in space. If you have not taken measures to account for this spatial clumping of the data, the resulting metrics of performance are likely to be unreliable and probably inflated. We need to see how these records are distributed in space to be able to assess whether the results are useful or not.

The reviewer is correct that transect sampling can lead to pseudo-replication and patterning at the scale of metres. Our description (in Table S1) of how the observations were collected and prepared for this analysis includes details of how we aggregated the various transect data to address this concern, which is already partially mitigated by the resolutions we used in our analysis. We also note that because our analysis is largely based on the relative comparison of different models, the question of inflated performance metrics is not relevant. 

To address the question of sampling distribution across the study area (as distinct from the pseudo-replication concern discussed above) we have added a supplemental figure (now S1) showing the spatial distribution of the sampling across the study area.

[55] Results in general: I would have found if much more useful and interpretable to have included both cross-validation and independent test scores in the same table, simplified down to just one or two example metrics: all the rest could go into the Supplementary Material. 

While we appreciate the volume of results presented is significant, our goal is not only to present the resulting substrate maps, but to examine ways the reliability of these maps can be assessed – something not previously considered in predictive models of substrate. In our view, this is essential to advance the usefulness of such models. This goal would be undermined by reducing the metrics presented to just one or two. 

We suggest that a comparison of cross-validation (Table 4) and independent data evaluation (Table 5) can be easily achieved by comparing individual columns in these two tables. No changes made. 

[56] Also, a question: where can the final map outputs be found? If the aim was to generate mapped predictions for use in environmental management, the outputs need to be accessible.

The maps will be available as georeferenced TIF files. We believe this will have higher utility to potential users than figures. We have, however, added the striking figure (showing the coastwide, 100 m classification) as an example of the results in the supplemental materials (Figure S5). 

[57] I found this section [Discussion] to read better than the others. I have not made detailed notes but I would make the same observation about the inferences around stationarity: given the imbalances in the spatial distribution and provenance of your sample and test data, can you really be sure that the differences in model performance you see among regions is attributable to non-stationarity in environmental process rather than artefacts in your input data?

First, there are no imbalances in the regional distribution or provenance of the build data used to develop or test the models (see new Figure S1 and the relevant supplementary tables on data provenance. And while we there are differences in sample density across depths, the models are not depth-specific so these differences cannot be causing any suck artefacts in the resulting models. 

In our view, the evidence for non-stationarity is overwhelming – starting with Fig. 2 which shows the differences in model performance across classes. If the same processes were at play in all regions, we would expect much more similar results across the regional models. This is supported by the highly variable importance of the different predictors in each region (Fig. 5). We note that this part of the analysis uses a coastwide data set, with no difference in spatial pattern across the regions (new Fig. S1). 

We suggest that this result is not, in fact, surprising, given that as we describe in our introduction and elsewhere it is now increasingly clear that stationarity is more of an exception than the rule. No changes made.

---

## [Decision Letter · Decision Letter 1]

9 Aug 2021

PONE-D-20-40989R1

Comprehensive marine substrate classification applied to Canada’s Pacific shelf

PLOS ONE

Dear Dr. Gregr,

Thank you for submitting your manuscript to PLOS ONE. It is obvious that you have met most of the reviewers' suggestions.  A final review does suggest some ways which the manuscript could be improved which we would like you to consider.

We look forward to receiving your revised manuscript.

Kind regards,

Judi Hewitt

Academic Editor

PLOS ONE

Journal Requirements:

Reviewers' comments:

Reviewer's Responses to Questions

**Comments to the Author**

1. If the authors have adequately addressed your comments raised in a previous round of review and you feel that this manuscript is now acceptable for publication, you may indicate that here to bypass the “Comments to the Author” section, enter your conflict of interest statement in the “Confidential to Editor” section, and submit your "Accept" recommendation.

Reviewer #2: (No Response)

2. Is the manuscript technically sound, and do the data support the conclusions?

Reviewer #2: Yes

3. Has the statistical analysis been performed appropriately and rigorously? 

Reviewer #2: Yes

4. Have the authors made all data underlying the findings in their manuscript fully available?

Reviewer #2: (No Response)

5. Is the manuscript presented in an intelligible fashion and written in standard English?

Reviewer #2: Yes

6. Review Comments to the Author

Reviewer #2: Thank you for addressing my earlier review comments and questions. I find the revised manuscript to have a much clearer logical flow and makes an interesting study more understandable. I still find the Introduction to be unnecessarily long, containing some material that I suggest could readily be condensed by referencing existing studies. The length of the Introduction is probably something for the editor to decide at this point but there are also a few points I find rather condescending, as currently worded. For instance, in the 'Model performance' section, I find dismissive generalisations such as "... metrics have evolved little ... with most studies continuing to report Cohen's Kappa ...", "adoption of improved metrics has been glacial ...", and "There is a persistent misconception about how to interpret model performance..." to be overly generalised (I am not a geologist but Kappa is very rarely used in relation to predictive model performance in the literature I am familiar with), didactic, and unnecessary.

I also still find the presentation of so many performance metrics to be more confusing than useful, for the most part. Indeed, while trying to interpret the results here I reflected that this illustrates one very good reason why a more refined set set of metrics is "commonly provided" in most published studies; practicality of interpretation. With the slightly revised focus in the title, however, (i.e., the method taking priority over the application) there is an argument for inclusion of more metrics.

A few minor comments:

116-117: As worded here, I don't see why it would follow that "higher resolution models would perform better in shallow waters" (which I interpreted as meaning that a high resolution model would work better in shallow water than it would in deeper water). I would guess the intended meaning might be worded as "higher resolution models would perform better than coarser resolution models in shallower waters."?

Line 226: "class weights" is unexplained, as yet, and therefore uninterpretable here.

Table 2: first column rows are not aligned with others? And "DEMs" is not defined.

Line 263: "cross-walked" is a term I've not seen before and only makes sense once you go to Table S1.

Table 3: Caption does not include "imbalance".

Line 474: Why "not unexpectedly"? If we think our models perform well, why would we not 'expect' them to perform equally well against independent data. Suggest there is no need for this in the sentence and, if retained, the expectation should be supported by references (there are a few recent papers on this subject).

Stationarity section: Now the input data are more fully described, particularly with the sample distribution map figure, this argument is better supported.

lines 590 onwards: Yes, I strongly agree with the points made in this section.

7. PLOS authors have the option to publish the peer review history of their article (what does this mean?). If published, this will include your full peer review and any attached files.

Reviewer #2: No

---

## [Author Response · Author response to Decision Letter 1]

15 Sep 2021

All comments are addressed in the uploaded response to reviewers.

---

## [Editor Report · Decision Letter 2]

14 Oct 2021

Comprehensive marine substrate classification applied to Canada’s Pacific shelf

PONE-D-20-40989R2

Dear Dr. Gregr,

We’re pleased to inform you that your manuscript has been judged scientifically suitable for publication and will be formally accepted for publication once it meets all outstanding technical requirements.

Kind regards,

Judi Hewitt

Academic Editor

PLOS ONE
---

## [Editor Report · Acceptance letter]

21 Oct 2021

PONE-D-20-40989R2 

Comprehensive marine substrate classification applied to Canada’s Pacific shelf 

Dear Dr. Gregr:

I'm pleased to inform you that your manuscript has been deemed suitable for publication in PLOS ONE. Congratulations! Your manuscript is now with our production department. 

Kind regards, 

on behalf of

Dr. Judi Hewitt 

Academic Editor

PLOS ONE